# What Is Known about Breast Cancer in Young Women?

**DOI:** 10.3390/cancers15061917

**Published:** 2023-03-22

**Authors:** Jie Wei Zhu, Parsa Charkhchi, Shadia Adekunte, Mohammad R. Akbari

**Affiliations:** 1Women’s College Research Institute, Women’s College Hospital, University of Toronto, Toronto, ON M5G 2C4, Canada; 2Department of Medicine, University of Toronto, Toronto, ON M5S 1A1, Canada; 3Institute of Medical Science, Faculty of Medicine, University of Toronto, Toronto, ON M5S 1A8, Canada; 4Dalla Lana School of Public Health, University of Toronto, Toronto, ON M5T 3M7, Canada

**Keywords:** breast cancer, young women

## Abstract

**Simple Summary:**

Breast cancer is the most common cancer affecting women under 40 years of age worldwide, with an increasing number of cases diagnosed each year. Despite this, breast cancer in young women is poorly understood as they are often underrepresented in clinical trials. Breast cancers in young women tend to be more aggressive and present at later stages as young women often do not meet screening age criteria. Recommended treatment may also be different due to unique fertility and psychosocial considerations. We herein summarize the unique challenges faced by young women, including risk factors, diagnosis, treatment, and survivorship issues, and draw attention to areas where further research is needed.

**Abstract:**

Breast cancer (BC) is the second leading cause of cancer-related death in women under the age of 40 years worldwide. In addition, the incidence of breast cancer in young women (BCYW) has been rising. Young women are not the focus of screening programs and BC in younger women tends to be diagnosed in more advanced stages. Such patients have worse clinical outcomes and treatment complications compared to older patients. BCYW has been associated with distinct tumour biology that confers a worse prognosis, including poor tumour differentiation, increased Ki-67 expression, and more hormone-receptor negative tumours compared to women >50 years of age. Pathogenic variants in cancer predisposition genes such as BRCA1/2 are more common in early-onset BC compared to late-onset BC. Despite all these differences, BCYW remains poorly understood with a gap in research regarding the risk factors, diagnosis, prognosis, and treatment. Age-specific clinical characteristics or outcomes data for young women are lacking, and most of the standard treatments used in this subpopulation currently are derived from older patients. More age-specific clinical data and treatment options are required. In this review, we discuss the epidemiology, clinicopathologic characteristics, outcomes, treatments, and special considerations of breast cancer in young women. We also underline future directions and highlight areas that require more attention in future studies.

## 1. Introduction

Breast cancer (BC) is the second most common cancer and the leading cause of cancer mortality for women, accounting for 685,000 deaths worldwide in 2020. Globally, breast cancer is responsible for one in four cancer cases and one in six cancer deaths in women [1]. Although more commonly diagnosed in women aged fifty years or older, the incidence of breast cancer in younger women is rising. It is currently the second leading cause of cancer-related mortality in women aged 0–39 worldwide, with 44,800 deaths per year. In Canada, approximately 1 in 200 women develop breast cancer by the age of 40 years, with a cumulative risk of breast cancer before 40 years of 0.61%, which is higher than the global average of 0.44% [2].

Breast cancer in young women (BCYW) is inconsistently defined. Previous studies have defined “young women” as age <35 years, dichotomized women into less than 40 years of age versus 40 and above, or included age as a proxy for menopausal status [3,4]. Consensus guidelines by the European School of Oncology and the European Society of Medical Oncology (ESMO) define ‘young women’ as women less than 40 years of age at the time of breast cancer diagnosis [5]. The consensus guidelines also distinguish young women from ‘very young women’, with the latter group comprising women of age ≤35 at diagnosis [6].

Young women constitute a special subpopulation of breast cancer patients with distinct tumour pathology, prognosis, diagnostic evaluation, treatment decision-making, survivorship, and fertility considerations. Compared to older women, tumours in young women tend to be more aggressive, with a higher proportion of estrogen receptor (ER) negative, triple-negative, and HER2+ tumours [3,4]. Younger age has also been associated with tumour biology that confers a worse prognosis, including poor tumour differentiation, increased Ki-67 expression, and more extensive lymph node involvement compared to women >50 years of age [5,7,8,9,10]. Further, a larger proportion of young women with breast cancer carry pathogenic variants in cancer predisposition genes such as *BRCA1/2*, compared to late-onset breast cancer [11,12,13]. The other important factor is that young women are often not included in breast cancer screening programs given their young age and are often diagnosed with more advanced disease. Several studies have found worse clinical outcomes and more long-term treatment complications among these young women compared to older breast cancer patients [14,15,16,17,18,19,20,21,22].

Breast cancer in younger women remains poorly understood. There is a lack of age-specific clinical characteristics or outcomes data for young women, and most of the standard treatments used in this subpopulation currently were tested in older patients. This review discusses the epidemiology, clinicopathologic characteristics, outcomes, treatment, and special considerations of breast cancer in young women.

## 2. Epidemiology

### 2.1. Geographic Variations

International variations in breast cancer incidence and mortality are poorly studied in young women compared to the general population. The global incidence of BCYW increased by 16% since the 1990s and breast cancer is currently the most common cancer in young women, with 244,000 cases diagnosed per year [2]. According to an analysis of GLOBOCAN 2018 data, the average risk of developing breast cancer by the age of 40 years is 0.44%. Across continents, the average cumulative risk of developing breast cancer before 40 years varied by approximately two-fold, being highest in Oceania (0.69%), followed by Europe (0.63%), the Americas (0.53%), Africa (0.49%), and lowest in Asia (0.38%) [2]. National comparisons across 185 countries showed that the highest cumulative incidence rate was in South Korea (0.95%), followed by United Kingdom, United States, and Canada (0.77%, 0.61%, and 0.61%, respectively), and lowest in Guinea (0.13%) [2]. Although Asia in general has the lowest average breast cancer incidence before age 40, there is an approximately six-fold difference among Asian countries [2]. These variations may be attributable to a lack of public registries with accurate population data and the increasing “westernization” of lifestyle habits among some of the developing countries (i.e., dietary changes and decreased physical activity) that increase breast cancer risk [23].

Breast cancer mortality for women aged <40 years varies worldwide despite similar incidence rates, suggesting a large disparity in case fatality rates of BCYW by geographic region. For example, although Western Africa and North America had similar age-standardized breast cancer incidence rates in women <40 years at 9.8 and 11.3 per 100,000, respectively, mortality rates differed greatly at 6.4 per 100,000 for Western Africa compared to 1.8 per 100,000 for North America [2]. Globally, the average risk of dying from breast cancer by 40 years of age was 0.08% in 2018. However, Africa had a notably higher risk than the world average at 0.18%. In 2018, worldwide mortality rates varied nearly 6-fold across regions, from 1.1 per 100,000 in Eastern Asia to 6.4 per 100,000 in Western Africa for women aged <45 years [2]. Differences in screening policies are less likely to account for this disparity, as young women aged <40 years generally do not qualify for breast cancer screening in any country. Therefore, other factors, including varying levels of awareness for breast cancer symptoms, time from diagnosis to treatment, and differences in treatment plans and accessibility to care may influence the observed differences, as lower income countries have lesser funding and capacity for cancer treatment [24].

Compared to developed countries, there is a higher breast cancer incidence rate among young women from low- and middle-income countries (LMIC), while the incidence rates show an increasing trend in LMICs. Regression analysis using data from Global Burden of Disease 2019 determined the age-standardized rates of incidence (ASIRs) and mortality in 60 countries from 2000 through 2019 and found that 21 countries showed an increasing incidence of breast cancer in women aged <40 years, while 16 countries demonstrated a decreasing trend [25]. Low HDI (Human Development Index) countries including Ecuador, Fiji, and Mauritius had the most significant increases, while high HDI countries such as Norway showed the largest decrease [25]. The remaining 23 countries showed a stable trend [25,26,27,28]. A population-based analysis of 645,000 premenopausal women diagnosed with breast cancer worldwide in 2018 found that the greatest burden of premenopausal breast cancer occurred in LMICs, with premenopausal (aged <50 years) breast cancer accounting for 55.2% of total breast cancer cases in low HDI countries, compared to 20.7% in very high HDI countries [26]. Although LMICs have a higher proportion of total breast cancer cases diagnosed at <50 years of age, the age-specific incidence rate of premenopausal breast cancer is higher in developed countries. Population analysis found the highest ASIRs for premenopausal breast cancer in Western Europe (38.4 per 100,000) and New Zealand (36.7 per 100,000), which were more than double that of south-central Asia (12.0 per 100,000) and eastern Africa (15.2 per 100,000) [26]. Risk factors contributing to the higher incidence of premenopausal breast cancer in developed countries are incompletely understood. Reproductive factors—fewer children, nulliparity, and childbearing at later ages—more common in developed countries are associated with earlier onset, usually hormone receptor-positive (HR+), breast cancer. Another possible explanation is the varied screening practices across geographical locations. Although North American and European guidelines recommend starting mammography at 50 years, earlier screening may be more accessible to young women in developed countries compared to those in LMICs [26]. In fact, 14% of women aged 18–39 years in the USA at average risk of breast cancer received a mammogram from 2011 to 2015 [27]. Keating et al. found that 19% of the BCs diagnosed during annual screening over 10 years are over-diagnosed and would not have become clinically apparent in the absence of screening [28]. Other observational studies have reported varying overdiagnosis rates ranging up to 54% [29]. Further research is warranted to better elucidate the genetic and environmental risk factors implicated in the incidence rate disparities and increasing incidence in both LMICs and higher income countries.

Compared to developed countries, there is a disproportionate burden of breast cancer incidence and mortality among young women from low- and middle-income countries (LMIC). The age-standardised mortality for premenopausal breast cancer in low HDI countries (8.5 cases per 100,000) was more than double the mortality in very high HDI countries (3.3 cases per 100,000) [26]. For example, although Canada and Nigeria have similar cumulative incidence rates of premenopausal breast cancer (0.61% and 0.64%, respectively), Nigeria had more than six times the cumulative mortality rate at 0.25% vs. 0.04% for Canada [2]. Although the cumulative risk of developing breast cancer by age 40 years is greater (0.67%) in high income countries compared to LMICs (0.34%), the case fatality ratio is almost four-fold higher in LMICs at 0.30% compared to 0.08% in high income countries [2]. Disparities in case-fatality rates for BCYW may be attributable to the treatment advances, early diagnosis, and starting mammography screening programmes earlier in high-income countries that substantially improve survival [30]. Other factors contributing to the current disparities in mortality rates are lower breast cancer awareness and sociocultural barriers to care for women in LMIC regions [24,25,26,27,28,29,31].

### 2.2. Ethnic/Racial Differences

Current evidence suggests that there are ethnic disparities in breast cancer incidence and mortality among young women. Black women aged <35 years have a higher breast cancer incidence rate than White women, although the overall age-adjusted breast cancer incidence rate is higher for White compared to Black women [32]. A retrospective analysis of nine SEER cancer registries between 1995 and 2004 found that the age-adjusted incidence rate for Black women aged <40 years was 16% higher compared to White women aged <40 years (incidence rate ratio = 1.16; 95% CI: 1.10–1.23) [33]. In addition, the age-adjusted mortality for Black women aged <40 years was more than twice the rate for White women age <40 (mortality rate ratio = 2.07; 95%CI: 1.99–2.14) [33]. Similarly, a recent analysis using data from the National Cancer Registration and Analysis Service that included 24,022 women aged 30–46 at the time of breast cancer diagnosis found that all ethnic minority groups apart from Indian women had a significantly greater odds of less favourable tumour characteristics compared to White women [34]. Multivariate analysis in the same study for women aged 30–46 years found that Black women had higher odds of having less favourable tumour characteristics compared to White women, including more advanced stage disease (OR = 1.58; 95%CI: 1.29–1.92), high grade disease (OR = 1.40; 95% CI: 1.18–1.66), and ER-negative disease (OR = 1.36; 95% CI: 1.09–1.70) [34]. The POSH prospective study of 2915 breast cancer patients aged 18–40 years also reported similar findings of higher median tumour diameter and higher frequency of triple negative tumours in Black compared to White women (26.1% vs. 18.6%, respectively, *p* = 0.04) [21,34]. Studies have suggested that lower surveillance attendance rates and increased prevalence of risk factors (i.e., parity and breastfeeding, higher BMI and hormone replacement therapy) in Black women compared to White women contribute to the less favourable tumour characteristics in young women; however, further studies are required [35,36,37,38].

Interestingly, the incidence of BC peaks at age 50 in Eastern and Southeastern Asia compared to 70 years in the United States [39]. The age-specific incidence of BCs in East Asian women aged 59 years and younger had a greater increase compared to US patients in recent decades. In the 40–49 age range, the probability of having an estrogen receptor positive (ER+) BC is significantly higher (OR = 1.50, 95% CI: 1.36–1.67, *p* < 0.001) although the probability of having a triple-negative breast cancer (TNBC) is lower compared to Americans (OR = 0.79, 95% CI: 0.71–0.88, *p* < 0.001) [40]. Additionally, the incidence rates of BC in some younger East Asian populations have surpassed those of the United States [41]. The increase in the incidence of BC in East Asian populations has been linked to dietary and reproductive factors that are representative of “westernization”. Such factors include high fat intake, low vegetable consumption, reduced parity, delayed childbearing, less breastfeeding, and late menopause [42]. Further, a body mass index (BMI) increase of 5 kg/m^2^ is associated with an increased risk for premenopausal BC in Asian women. However, in African and Caucasian women, an inverse association has been observed [43].

In a study of women with BC aged 20–49 years, Black, American Indian or Alaska Native, and Hispanic women had a greater frequency of diagnosis in late stages compared to White and Asian or Pacific Islander women [44]. Additionally, a higher proportion of African American and Hispanic BC patients aged 15–39 experience delayed treatment after diagnosis compared to White Americans (*p* < 0.001). Further analysis showed that longer treatment delay time was a risk factor for shorter survival (*p* < 0.001) [45].

### 2.3. Risk Factors

Several risk factors are implicated in developing breast cancer in young women (Figure 1). Factors associated with BC development are classified into lifestyle risk factors (i.e., physical activity, body habitus, and alcohol consumption), inherent or genetic risk factors, reproductive risk factors, and iatrogenic risk factors.

#### 2.3.1. Lifestyle Risk Factors

Lifestyle risk factors include physical activity, body mass index (BMI), alcohol, smoking, socioeconomic status, and certain occupational conditions. The current evidence suggests that physical activity is associated with a dose-dependent reduction in the risk of early onset breast cancer for all types of activity and should be recommended. Previously, the consensus was that premenopausal BC risk is independent of physical activity levels, following several prospective cohort studies that reported no association [46,47,48,49,50,51,52,53,54,55,56,57,58,59,60,61,62,63,64,65,66,67,68,69,70,71]. For example, a large prospective cohort study by Rockhill et al. examining the physical activity levels of 104,468 women over a 6-year follow-up period found that women who were more physically active (i.e., engaged in a strenuous activity at least twice per week for 10–12 months per year in late adolescence) did not have a lower risk of developing breast cancer compared to those who did not engage in physical activity (RR = 1.1; 95% CI: 0.8–1.6) [49]. However, recent data since then have contradicted these findings, with three meta-analyses concluding that physical activity significantly reduced the risk of developing premenopausal breast cancer [72,73,74]. In a meta-analysis of 6 studies that included 2258 cases, Wu et al. compared women in the highest versus lowest categories of physical activity and found that increased physical activity was inversely associated with breast cancer risk and a 23% reduction in breast cancer cases (RR = 0.77; 95%CI: 0.72–0.84) [72]. Similarly, Hardefelt et al. conducted a meta-analysis of 48 cohort studies with 236,955 breast cancer cases and 3,963,367 controls, and concluded that physical activity significantly reduced the overall risk (OR = 0.79; 95% CI: 0.73–0.87) [73]. Chen et al. examined 38 cohort studies with 68,416 breast cancer cases and found a reduced risk of developing premenopausal breast cancer with physical activity (RR = 0.83; 95% CI: 0.79–0.87) [74]. The main mechanism by which physical activity acts as primary prevention for breast cancer development is by lowering the cumulative exposure to circulating ovarian hormones [75,76,77]. Specifically, strenuous physical activity at the pre-pubertal stage delays the onset of regular ovulatory cycles, while activity during reproductive years reduces circulating ovarian hormone levels, frequency of regular cycles, and fat stores where androgen is converted to estrone [75]. Although emerging evidence suggests that increased intensity and duration of exercise are associated with lower breast cancer risk [78], it remains unclear whether specific types of physical activity (e.g., recreational, occupational, or non-occupational) are more strongly associated with a reduced risk of breast cancer in young women.

The effect of BMI on the risk of developing breast cancer differs between pre- and post-menopausal women. In post-menopausal women, there is a positive correlation between increasing BMI and breast cancer risk. In contrast, several studies have demonstrated that there is a modest protective effect of increased BMI that is inversely associated with the risk for developing breast cancer in young women [43,79,80,81,82,83,84,85]. Renehan et al. conducted a meta-analysis with 20 prospective cohort studies and found a dose–response effect with every 5 kg/m^2^ increase in BMI associated with a significant decrease in the relative risk (RR) of developing breast cancer in premenopausal women (RR = 0.92; 95% CI: 0.88–0.97) [79]. The underlying mechanism is unknown; it is hypothesized that obesity may lead to ovarian suppression and lower levels of circulating estradiol [86]. Although higher BMI is protective in young women, several studies have found that the risk reduction is offset by the larger cumulative post-menopausal risk for developing breast cancer later in life [79,87,88,89,90]. Interestingly, a meta-analysis by Amadou et al. that included nine case-control studies and three cohort studies found a significant dose–response increase in the relative risk of premenopausal BC (RR = 1.08; 95% CI:1.01–1.16) per 0.1 unit increase in waist-to-hip ratio (WHR), even though a simultaneous increase in BMI by 5 kg/m^2^ was associated with a significantly decreased risk (RR = 0.95; 95%CI: 0.94–0.97) [43]. This suggests that high general adiposity (indicated by BMI) reduces risk, while central adiposity (indicated by WHR) is conversely associated with an increased risk of premenopausal breast cancer. The Carolina Breast Study also found that BMI was inversely correlated with premenopausal breast cancer risk in White but not Black women [91], while a higher WHR adjusted for BMI was associated with increased breast cancer risk in both Black and White premenopausal women [92]. Overall, given that obesity is associated with increased risk for developing other malignancies and health complications, weight gain is not recommended as a method for breast cancer risk prevention.

Many studies examining the effect of alcohol consumption have found an increased risk of developing breast cancer in young women. In 1997, the first study examining the impact of alcohol intake among young women (defined as <45 years of age) by Swanson et al. found that women who drank >14 alcoholic beverages per week were more likely to develop breast cancer than non-drinkers (RR = 1.73; 95% CI: 1.2–2.6) [93]. Since then, several studies reported similar findings [94,95,96]. A pooled multivariate analysis of 3730 premenopausal women suggested a dose–response effect of alcohol on breast cancer and found that an incremental increase in alcohol consumption by 10g per day was associated with an increased risk of breast cancer (RR = 1.03; 95% CI: 0.99–1.08) [97]. Similarly, a meta-analysis conducted in 2018 by the World Cancer Research Fund also found a statistically significant elevated risk for developing breast cancer in premenopausal women, with a 5% increased risk for every 10 g increase in ethanol per day [98]. Interestingly, the same analysis found that different types of alcohol had varying effects on the risk for premenopausal breast cancer, with beer having the highest risk (RR = 1.32; 95%CI: 1.06–1.64) and spirits having the lowest (RR = 1.10; 95%CI: 0.92–1.30) [98].

The role of active smoking in BCYW risk has remained unclear since it was first discussed in 1982 [99,100]. Most studies concur that if smoking is implicated in breast cancer risk, it likely plays a more significant role in premenopausal compared to post-menopausal women [101,102]. Earlier age of smoking initiation appears to have a higher lifetime risk for breast cancer than those who start smoking later in life. Jones et al. conducted a cohort study with 1815 invasive breast cancer cases and found that the hazard ratio for women who started smoking before the age of 17 was 1.24 (95%CI: 1.08–1.43; *p* = 0.002) compared to non-smokers, which was higher than all “ever” smokers (HR = 1.14; 95%CI: 1.03–1.25) [103]. However, a meta-analysis demonstrated that the effect of smoking on breast cancer risk is confounded by its close association with alcohol consumption [100]. Other studies have also suggested that passive smoking may pose more risk for BCYW than active smoking. Active smoking is hypothesized to exert an anti-estrogenic effect that counteracts the risk associated with exposure to smoking-related carcinogens. In contrast, passive smoking concurs the same risk associated with carcinogen exposure over time without the protective anti-estrogenic effect [102,104,105]. A meta-analysis of 14 studies found that passive smoking was associated with an increased risk (pooled RR = 1.68; 95%CI: 1.88–2.12) [101]. Another study found that passive smoking increased the risk of premenopausal BC in carriers of *PARP1* or *ESR1* mutations (OR = 1.54; 95%CI: 1.14–2.07) [106]; however, further studies are required to elucidate individual risk levels stratified according to genetic susceptibility.

Data on the impact of socioeconomic status (SES) on population-level breast cancer risk are currently limited. A study by Akinyemiju et al. examining the SES of different ethnicities within a U.S. population found that the combined risk of both early and late breast cancer increased with higher SES [106]. Current evidence suggests that although women of higher SES in childhood or born to a mother with higher educational attainment are at higher risk of developing breast cancer, survival outcomes are better compared to those with lower income [107,108]. One possible explanation for these findings is that individuals from higher SES tend to be older at the time of their first pregnancy and have lower parity compared to those from lower SES [108].

Occupation-related long-term night shifts in young adulthood may be another factor contributing to the increased risk of developing breast cancer among young women. It is hypothesized that working overnight causes circadian rhythm disruption, as the “light at night” induces the suppression of melatonin production from the pineal gland [109]. Pre-clinical trials have demonstrated that melatonin is associated with tumour-suppressive effects through several mechanisms, including modulating estrogen production and exerting an anti-estrogenic effect [110]. As such, dysregulated or decreased melatonin production may promote tumour growth. According to an analysis of the Nurses’ Health Study II that included 116,430 female registered nurses aged 25–42 years with a 24-year follow-up, the risk of developing breast cancer was significantly elevated among those who had ≥20 years of cumulative rotating night shift work (HR = 1.40; 95%CI: 1.00–1.97) compared to those who did not [111]. These findings are supported by another case-control study that found night shift work was associated with higher breast cancer risk among pre-menopausal (OR = 1.33; 95%CI: 0.98–1.79) than post-menopausal women (OR 1.08; 95%CI: 0.82–1.42) [112]. More recent analyses have also reported that the odds ratio of developing pre-menopausal breast cancer was 1.26 (95% CI:1.06–1.51) for women who had ever worked night shifts compared to those who did not, and that the risk increased with both increasing frequency and number of years with night shift work [113,114].

#### 2.3.2. Genetic Risk Factors

International guidelines recommend that patients younger than 50 years of age or with TNBC should be referred for genetic counselling. Common clinical features suggestive of hereditary breast cancer include high cancer incidence or the same type of cancer within a family, early age of onset (<50 years of age), different cancers in a person, bilateral disease, and multifocality. Timely identification of genetic mutations and screening of family members is crucial for prevention and treatment to improve long-term outcomes. Recent efforts have analyzed the mutational landscape of hereditary breast cancer using next-generation sequencing (NGS) and microarray genotyping. Predisposing mutations may be categorized based on their relative risk for developing a specific type of cancer as high-penetrant mutations (associated with cancer RR > 5) or moderate-penetrant mutations (RR = 1.5–5) and low-penetrant loci (RR < 1.5), with varying risks between different populations and age groups (Table 1) [115]. Approximately 5–10% of breast cancer cases are hereditary, of which 50% are estimated to be caused by deleterious mutations in high or moderate penetrance genes [116,117].

##### High-Penetrance Genes

High-penetrance genes include BRCA1, BRCA2, TP53, PTEN, STK11, CDH1, and PALB2 [120].

To date, BRCA1 and BRCA2 are the most common and widely studied breast cancer susceptibility genes, accounting for up to 40% of familial breast cancer. The BRCA1 gene codes for a nuclear phosphoprotein involved in DNA damage response, cell cycle progression, centrosome number, and regulating transcription, while BRCA2 encodes a protein responsible for double-stranded break repair during homologous recombination (HR) [121]. Unlike BRCA1-associated cancers, BRCA2 tumours often express estrogen and progesterone receptors with similar features as sporadic breast cancers. In a pooled analysis, the risk for breast cancer before the age of 40 years in carriers of BRCA1/2 germline mutations was 9.4% to 12% [120]. In an analysis of two population-based case-control studies involving 2013 women diagnosed with breast cancer before age 35 years and no family history and 225 affected women under the age of 45 years and a positive first-degree family history of breast cancer, Malone et al. found that 12% of women aged <45 years with a family history of breast cancer were carriers of BRCA1/2 mutations. The same study found that among women aged <35 years diagnosed with breast cancer and no family history, 9.4% (7.1% for BRCA1 and 4.9% for BRCA2) were carriers of germline mutations [12]. These results are consistent with a prospective cohort study by Copson et al. which found that 12% (338 of 2733) of breast cancer patients aged <40 years were carriers of germline BRCA1/2 mutations [122]. A recent study by the Breast Cancer Association Consortium that examined the risk of breast cancer for protein-truncating germline variants in nine genes found the highest breast cancer risk for age <40 years associated with BRCA1 (OR = 32.8; 95% CI: 16.9–63.4) and BRCA2 (OR = 11.9; 95% CI: 7.33–19.4) [118]. Similarly, the CARRIERS population-based study examined pathogenic variants in predisposition genes and risk of breast cancer in women < 45 years of age and reported the strongest association between BRCA1 (OR = 8.63; 95% CI: 5.63–13.89; *p* < 0.001) and BRCA2 (OR = 7.65; 95%CI: 5.47–11.02; *p* < 0.001), and the risk of developing breast cancer before the age of 45 years (Table 1) [119].

TP53 mutations are highly penetrant and associated with several cancers, of which early-onset breast cancer is the most common tumour type among women with germline TP53 mutations [123]. The protein encoded by the TP53 gene responds to cellular stress and is implicated in regulating the expression of genes in different pathways, inducing cell cycle arrest, senescence, apoptosis, and DNA repair [123]. The lifetime risk for developing breast cancer in women who are TP53 mutation carriers is approximately 50% [124]. Among women <35 years of age diagnosed with breast cancer, the frequency of germline TP53 mutations ranges from 1–7% and can reach up to 30% in those diagnosed before the age of 30 [118,124,125,126,127,128,129].

PTEN encodes a protein that suppresses the P13K/Akt/mTOR pathway and regulates cell metabolism, proliferation, and survival [130,131]. Germline pathogenic PTEN mutations are uncommon and associated with a constellation of clinical manifestations, including Cowden syndrome [132]. Breast cancer is the most commonly diagnosed malignancy among women with Cowden syndrome, with a lifetime risk of 85%, typically associated with thyroid and endometrial tumours, and the age of diagnosis between 38 and 46 years [133,134]. Studies have demonstrated that the frequency of PTEN mutations in women <40 years of age is <1% and estimated that the risk of developing breast cancer for all ages is 2- to 5-fold higher among PTEN mutations carriers compared to noncarriers [133,134,135].

STK11/LKBI encodes a tumour suppressor protein involved in cell metabolism, proliferation, and p53-dependent apoptosis. Pathogenic STK11 mutations increase susceptibility to breast, pancreatic, and gastrointestinal cancers, with a cumulative incidence of 45% [135]. However, Hearl et al. found an only 8% risk of breast cancer by the age of 40 among women with STK11/LKB1 pathogenic variants, with a non-significant difference (log-rank test of difference = 0.62; *p* = 0.43) between carriers of STK11/LKB1 mutations compared to non-carriers [136].

Loss of function mutations in CDH1 that encodes E-cadherin are associated with increased cell proliferation, invasion, and metastasis. CDH1 germline mutations are associated with autosomal dominant hereditary diffuse gastric carcinoma (HDGC), of which approximately 30% of patients present with invasive lobular breast cancer [137,138,139]. Female carriers of pathogenic CDH1 mutations have a 40–54% lifetime risk of developing lobular breast cancer, typically presenting with an early-onset disease with a mean age at diagnosis of 40 years and bilateral breast cancer [124,140,141,142,143].

PALB2 encodes a tumour suppressor protein that recruits BRCA2 to DNA damage sites. Heterozygous germline PALB2 mutations are associated with an increased risk for breast cancer, with previous studies reporting a higher penetrance in younger than older women. Antoniou et al. analyzed the risk of breast cancer among 362 women with identified deleterious PALB2 mutations and found that the risk of developing breast cancer was 8- to 9-fold higher in PALB2 mutation carriers who were <40 years of age and 5- to 8-fold higher in those >40 years of age compared to the general population [144]. The same study reported that the cumulative breast cancer risk among PALB2 mutation carriers was 14% (95% CI: 9–20) by 50 years of age [144]. Further, the odds ratio for PALB2 was reported to be 5.36 (95% CI: 2.26–12.7) for the <40 age group [118].

##### Moderate-Penetrance Genes

Moderate penetrance genes include CHEK2, ATM, BRIP1, BARD1, RAD51C, and RAD51D, which account for approximately 5% of the hereditary risk [120].

CHEK2 encodes a G2 checkpoint kinase that responds to DNA damage and prevents mitosis. A meta-analysis by Weischer et al. with 26,000 cases and 27,000 controls found that heterozygotes of CHEK2*1100delC had an OR of 2.6 (95% CI, 1.3–5.5) for developing early-onset BC compared to non-carriers [145]. The same study found that approximately 0.64% of all early-onset cases (<51 years) were heterozygous for CHEK2*1100delC, premenopausal and with ER+ disease. Several studies have identified CHEK2 mutations in specific ethnic populations. Rashid et al. assessed the prevalence of CHEK2 germline mutations in 145 BRCA1/2-negative breast cancer patients from Pakistan and found a low frequency (1.4%) of two potentially deleterious missense mutations, c.275C>G (p.P92R) and c.1216C>T (p.R406C) in women aged <40 years [146]. In another study, the CHEK2 Y390C (1169A>G) mutation was found in 8% (12 of 150) of young Chinese women with BC aged <35 years that were significantly higher than controls [147].

The ATM gene encodes a PI3/PI4-kinase involved in cell cycle checkpoint signaling pathways that are sensitive to DNA damage and it regulates p53, BRCA1, and RAD17, among others. Thompson et al. conducted a prospective study involving 247 heterozygous carriers of ATM mutations and found a higher relative risk for developing breast cancer in carriers <50 years of age (RR = 4.94; 95%CI: 1.9–12.9) compared to the general population [148]. In another study by Maillet et al., which included 94 patients <40 years of age diagnosed with breast cancer without any family history of breast cancer and 140 healthy controls, they identified germline ATM variants among 10 breast cancer cases (10.6%, 95%CI: 5.2–18.7%) with no mutation carrier found in the control group (*p* = 0.0006) [149]. Similarly, Teraoka et al. analyzed genomic DNA samples of women diagnosed with breast cancer before 45 years of age compared to matched controls, and detected ATM mutations among 11 of the 142 breast cancer cases (7.7%; 95% CI, 3.9 –13.4%) compared to 1 of 81 controls (1.2%; 95% CI, 0.0–6.7%) (*p* = 0.06) [150]. In this study, all the cases with identified mutations had a first-degree family history of breast cancer (OR:12.1; 95% CI, 6.2–20.6, *p* = 0.02). Several other studies have reported similar findings, with ATM mutations identified in women <45 years of age and a higher frequency in cases with a positive family history of breast cancer [151,152,153,154,155].

BRIP1 mutations are associated with breast and ovarian cancers and encode a protein that interacts with BRCA1 and is involved in double-stranded DNA break repair. Couch et al. analyzed germline DNA samples from 1824 cases of TNBC and identified BRIP1 mutations in 8 patients with a mean age of diagnosis of 46 years (range 36–68), suggesting that BRIP1 mutations may be more common in women with earlier onset and TNBC [156]. Other studies have also identified BRIP1 mutations (c.2392C>T) in an Irish cohort comprising patients with breast cancer diagnosed at <42 years of age and the p.Q994E mutation in Chinese patients with early-onset breast cancer diagnosed at <35 years of age [157,158]. Overall, studies have estimated that there is a 1.2- to 3.2-fold higher risk for developing breast cancer in patients <40 years of age with identified BRIP1 mutations compared to non-carriers, and that 1% of patients with early-onset or familial breast cancer carried a deleterious BRIP1 mutation [152,153,156].

The BARD1 protein interacts with BRCA1 and is thought to play a critical role in BRCA1-mediated tumour suppression [159]. Studies have reported polymorphic variants associated with a 2.5-fold increased risk for developing early-onset breast cancer [155,160,161]. In addition, the p.Cys557Ser variant was first identified in the analysis of fresh frozen tissue in 5 women with BRCA-negative breast cancer within an Italian family and was absent in controls [162], and later found to have a frequency of 7.4% in 94 Finish breast cancer patients whose family history did not include ovarian cancer [163]. The c.Cys557Ser mutant allele had a 2.8% frequency in an Icelandic population with breast cancer compared to 1.6% in controls (OR = 1.82; 95%CI: 1.11–3.01; *p* = 0.014) [159]. In this study, the two patients who were homozygous for the p.Cys557Ser variant were diagnosed with breast cancer at the ages of 41 and 47 [159].

The RAD51 family (RAD51B, RAD51C, RAD51D, XRCC2, and XRCC3) transduces the DNA damage signal and is a critical protein in homologous recombination through p53 interactions. In an association analysis that used a panel of 34 putative susceptibility genes to perform sequencing on samples from 60,466 women with breast cancer and 53,461 controls, pathogenic germline variants in RAD51C (OR = 4.83; 95% CI: 0.52–45.2) and RAD51D (OR = 1.76; 95% CI: 0.38–8.17) were both found to be associated with the risk of developing breast cancer before 40 years of age [118]. Another study analyzed data from 7216 families, among which 215 women had RAD51C pathogenic variants and 92 women had RAD51D pathogenic variants. For RAD51C, a higher BC relative risk in younger women aged 20–49 years (RR = 2.42; 95% CI: 1.61–3.63) was observed compared with older women ≥50 years (RR = 1.36, 95% CI: 0.70–2.63). For RAD51D, the RR for the 20–49 age group was 1.84 (95% CI: 1.12–3.02); whereas, the RR for the 50–79 age group was 1.83 (95% CI: 1.02–3.26) [164]. The estimated cumulative risks of developing BC by age 50 years were 4% (95% CI = 3% to 6%) for carriers of pathogenic variants in RAD51C and 4% (95% CI = 2% to 5%) for RAD51D carriers [164].

##### Low-Penetrance Genes

Over 180 single nucleotide polymorphisms (SNPs) identified by genome wide association studies (GWAS) are considered low penetrance for breast cancer and account for 18% of the hereditary risk [120]. These studies involving large samples of cases and controls have allowed the identification and assessment of low-risk loci and led to the development of a polygenic model for breast cancer [165,166,167]. However, the relevance of polygenic risk score to BCYW has not been studied yet.

#### 2.3.3. Reproductive Risk Factors

Several hormonal and reproductive factors, including hormonal contraceptives, hormonal therapy, pregnancy, and breastfeeding, may influence the risk for BCYW (Table 2). There has been an increased usage of exogenous hormonal medications, which include oral contraceptive pills (OCPs), intrauterine devices (IUDs), and menopausal hormonal therapy (MHT). In the past few decades, there has also been a shift within high-income countries towards having fewer children per household and pregnancies later in life.

In 2018, OCPs were the most common contraception method used in women aged 15 to 49 [173]. In 1996, a large dataset study first suggested an association between combined OCP (i.e., estrogen and progesterone preparations) and increased breast cancer risk. The analysis included 53,297 breast cancer cases and 1,000,239 controls and found a 1.24-fold elevated risk for breast cancer in combined OCP users (RR = 1.24; 95%CI: 1.15–1.33) compared to controls [174]. For every 20,0000 women aged 20–25 years using OCPs, there is one woman who would develop breast cancer. The study also found that the modestly elevated risk resolved after 10 years (RR = 1.01; 95% CI: 0.96–1.05) [174]. Data for OCP use in high-risk women who have genetic risk factors or positive family history are limited, but the current evidence suggest a similar effect of OCP use on BC risk compared to the overall population [175]. For high-risk individuals with *BRCA1* or *BRCA2* mutations considering combined OCPs, genetics consultation should be considered to judge the competing impacts of increased breast cancer risk against individual needs and potential protective effects against ovarian cancer [175]. In a prospective cohort study by Mørch et al. that examined the association between breast cancer and OCP use in 1,797,932 Danish women aged 15 to 49 years over a mean follow-up period of 11 years, the authors found a higher risk of breast cancer among those who currently or recently used combined OCPs (RR = 1.20 (95%CI: 1.14 to 1.26) and this increased with duration of use, from 1.09 (95% CI, 0.96 to 1.23) with <1 year of use to 1.38 (95% CI, 1.26 to 1.51) after more than 10 years of use (*p* = 0.002) [176]. Few studies have examined progestin-only contraceptives and breast cancer risk. One study analyzing a cohort of 93,843 women who used the levonorgestrel-releasing intrauterine system (LNG-IUS) reported a modestly elevated breast cancer risk (RR = 1.19; 95%CI: 1.13–1.25) compared to the general incidence rate among Finnish women younger than 55 years of age [177]. Similarly, a recent systematic review and meta-analysis found an overall increased breast cancer risk (OR = 1.16; 95%CI: 1.06–1.28) in LNG-IUS users aged <50 years, which increased in women over 50 years (OR = 1.52 (95%CI: 1.34–1.72) [171]. However, considering the potential benefits of OCP use with long-term follow-up data on women using the combined OCP suggesting protective effects against ovarian, endometrial, and colorectal cancer, and the clinical indications for progesterone OCPs in controlling menorrhagia, risk–benefit counselling should be provided on an individual basis before commencing hormonal contraception [178].

Premature menopause, defined as occurring <40 years of age without iatrogenic causes, is rare and affects approximately 1% of women. Menopausal hormonal therapy (MHT) is often recommended in this population for symptom relief or bone protection, and may contain either unopposed estrogen or combined estrogen and progestin preparations. In 2019, the Collaborative Group on Hormonal Factors in Breast Cancer conducted a large meta-analysis of international epidemiological evidence for the type and timing of MHT associated with breast cancer risk. This study reported that the use of MHT for 5 to 15 years, starting between the ages of 30–39 years, was not associated with a statistically significant increased risk (RR = 1.07; 95% CI: 0.88–1.31) of breast cancer [37]. Current guidelines recommend that average-risk women with early menopause can be offered either combined HRT if their uterus is intact or estrogen-only HRT if their uterus has been removed up to the natural age of menopause (typically 51–52 years), but more caution should be exercised with the lowest dose and shortest duration, and specialist input for high-risk patients [179].

Age of pregnancy has been shown to influence breast cancer risk. In developed countries, approximately 13% of breast cancers are diagnosed during the reproductive years between 20 to 44 years of age, and about 2% of women are diagnosed before 35 years of age [180]. Parity before 20 years of age is associated with the largest long-term risk reduction of 50% compared to nulliparous women [180], while being >35 years of age at the time of first pregnancy is associated with increased breast cancer risk (incidence RR = 1.18; 95%CI: 1.04–1.34) compared to nulliparous women [181]. However, there is a transiently increased breast cancer risk post-partum that is attributed to the involution process within the breast. Breast remodeling is hypothesized to occur after delivery due to the lactational changes in the post-partum period, and the increased risk associated with immune micro-environmental changes may last for up to 10 years [182]. Following this period, the lower breast cancer risk is likely due to a reduction in the ER-sensitive epithelial cells in breast tissue [182]. Parous women have an increased breast cancer risk peaking 5 years after delivery (HR = 1.80; 95%CI: 1.63–1.99) and this decreases to 0.77 (95%CI: 0.67–0.88) 34 years later [183]. The 1993–2001 Carolina Breast Cancer Study included 1809 White and 1505 African American women and found important racial differences in the effect of parity on breast cancer risk in young women between 20 to 49 years of age [184]. Among young African American women, multiparity was associated with increased breast cancer risk (OR for 3–4 pregnancies: = 1.5, 95%CI: 0.9–2.6; OR for ≥5 pregnancies = 1.4; 95% CI: 0.6–3.1) [184]. In contrast, White women did not appear to have increased risk (OR for 3–4 pregnancies = 0.7; 95% CI: 0.4–1.2 and OR for ≥5 pregnancies = 0.8, 95%CI: 0.2–3.0) [184]. A prospective cohort study examining 7152 Chinese women with primary breast cancer also found that younger patients aged <40 years were more likely to be nulliparous compared to patients ≥40 years of age (43.3% vs. 17.8%; *p* < 0.001) [22]. Future studies are necessary for detailed comparisons of breast cancer risk in young women between different racial and ethnic groups.

Women who have children later in life may opt for fertility techniques such as oocyte harvesting, oocyte cryopreservation, embryo transfer, or in vitro fertilization (IVF). No association (HR = 0.79; 95%CI: 0.46–1.36) has been found between fertility preservation techniques and increased breast cancer risk, including in high-risk patients with *BRCA1/2* mutations [185]. However, a systematic review found limited evidence on the effect of IVF on breast cancer in premenopausal women and further research is required [186].

Breastfeeding has been reported to reduce breast cancer risk [187]. Although the risk reduction mechanism is currently unclear, the relative risk reduction of 4% for every 12 months of breastfeeding for women of all ages and a greater reduction of 5.1% for premenopausal breast cancer have been reported [188,189]. Studies have found a similar protective effect of breastfeeding for hormone receptor-positive and negative breast cancer subtypes; although, there was a stronger inverse association reported for TNBC (OR: 0.78; 95% CI: 0.66–0.91) which is more common in younger women [190,191]. According to the World Health Organization recommendations, young mothers should be supported to ensure breastfeeding is continued for at least six months before weaning to benefit from its protective effect [192].

## 3. Prognosis

The current evidence suggests that breast cancer prognosis is worse for younger women. Several factors contribute to the poorer prognosis, including a higher proportion of cases presenting with more advanced stages, aggressive clinicopathological features, and increased risk of cancer recurrence and mortality (Figure 2) [193,194,195].

### 3.1. Clinicopathological Characteristics

Studies have consistently found that breast cancers in young women have more aggressive clinicopathological features with larger tumours, higher grade, more lymph node positivity, lower hormone receptor positivity, higher HER2 overexpression, and a higher proportion of triple-negative cases when compared to older women. A prospective study by Colleoni et al. that included 1427 women <50 years of age found a larger proportion of ER- (38.8% vs. 21.6%; *p* < 0.001), PR- (49.1% vs. 35.3%, *p* < 0.001), vascular or lymphatic invasion (48.6% vs. 37.3%; *p* = 0.006), and pathologic grade 3 tumours (61.9% vs. 37.4%; *p* < 0.0001) in women aged <35 years compared to those aged 35–50 years, respectively [196]. More recently, another large prospective study evaluating 2956 breast cancer patients aged <40 years at diagnosis reported that most women had ductal histology (86.5%) and grade III (58.9%) disease [21]. This study found that 50% of women had node-positive disease, and 27% had multifocal tumours. Approximately one-third (34%) of cases were ER- tumours, while one-quarter (24%) were HER2+ [21]. Similarly, the Young Women’s Breast Cancer study that included 399 women aged ≤40 years at diagnosis also reported the highest proportion of grade 3 breast cancer in the youngest subgroup (≤30 years = 64%, 31–35 years = 57%, and 36–40 years = 53%), high rates of lymphovascular invasion (34%), and lymphocytic infiltration (24%) [196]. This study found that 11% of cases were HER2 positive, and 21% had TNBC. Other retrospective studies have also assessed the differences in breast cancer in different age groups. The largest analysis conducted by Gnerlich et al. included >200,000 women with breast cancer, of whom approximately 15,000 cases were diagnosed <40 years of age. The authors reported that young women were more frequently diagnosed with higher grade, larger size (>2 cm), lymph-node positive, poor differentiation, and ER-/PR- tumours (*p* < 0.0001). Further, a California Cancer Registry study examining 5600 women reported a statistically significant increased odds of being diagnosed with advanced-stage (stage III or IV) disease in women aged <40 years at diagnosis (OR = 1.33; 95% CI: 1.24–1.43) compared to women >40 years [197]. The proportion of TNBC was also highest in women <40 years of age (22.8%) and decreased with age increase (14.3% for 40–49 years and 11.7% for >50 years). Racial differences also exist in TNBC prevalence, which has been reported to be highest at 56% in African Americans and 42% in White women aged 20–34 years [198].

Genomic analyses can examine the unique biology of breast cancer in young women. Clinically annotated microarray data from 784 early-stage breast cancers were collected for two age-based cohorts for women ≤45 years and ≥65 years of age. Genomic expression analysis showed unique biological features with significantly lower mRNA levels of ERα (*p* < 0.0001), ERβ (*p* = 0.02), and PR (*p* < 0.0001), and higher mRNA levels of both HER2 (*p* < 0.0001) and epithelial growth factor receptor (EGFR) (*p* < 0.0001) among women ≤45 years [198]. Importantly, an exploratory analysis with Gene Set Enrichment Analysis (GSEA) identified 367 significant genes enriched among tumours in young women only [198]. These genes comprised various functions, including immune response, hypoxia, BRCA1, apoptosis, and several oncogenic signalling pathways (i.e., mammalian target of rapamycin (mTOR), Myc, E2F, and Ras) [198]. In addition, differentially expressed genes in normal mammary tissues and tumours may also be influenced by the menstrual cycle in premenopausal women. Using next-generation sequencing (NGS), Pardo et al. examined transcriptome changes as a function of the luteal and follicular phases of the menstrual cycle from 20 normal breast tissue samples [199]. There were 221 overexpressed and 34 downregulated genes identified during the luteal phase compared to the follicular phase in this study. All studies to date support the notion of unique genomic alterations in BCYW being distinct from those in older women, underlying the more aggressive phenotype.

### 3.2. Survival

Younger age has been shown to be an independent poor prognostic factor for breast cancer patients [5,39,200,201,202]. An early retrospective analysis of more than 1200 women diagnosed with early-stage breast cancer found that age younger than 35 was a strong prognostic factor in multivariate analyses for time to cancer recurrence (RR = 1.70, *p* < 0.001) and overall mortality (RR = 1.50, *p* < 0.04) [203]. This analysis suggested that young age, after adjusting for all known prognostic factors, may be a predictor of recurrence risk and survival. Another retrospective study that analyzed 200,000 women in the SEER database diagnosed with breast cancer found that women aged <40 years were 39% more likely to die when compared to those aged 40 or older (HR = 1.39; 95% CI: 1.34–1.45) [204]. The 5-year breast cancer-specific survival increased from 74.0% to 88.5% between 1975 and 1979 and 2010 and 2015 for women <40 years of age diagnosed with breast cancer [201]. Between 1988 and 2003, there was a significantly higher mortality rate in younger women <40 years of age compared to those >40 years (18.3% vs. 12.1%; *p* = 0.001), and the largest disparity was observed for early-stage disease [204]. Women aged <40 years were 44% (HR = 1.44; 95% CI: 1.27–1.64) and 9% (HR = 1.09; 95% CI: 1.03–1.15) more likely to die of stage I and stage II breast cancer, respectively [204]. Further, a significantly lower overall survival (OS) and disease-free survival (DFS) have been reported in Mexican, Hungarian, and Indian cohorts in very young (<35 years of age) women compared to young (<45 years), which may be related to the more aggressive subtypes of tumours developing in earlier ages [205,206,207]. Bajpai et al. conducted a prospective cohort study enrolling 1228 women aged ≤40 years. Although this study did not find significant difference in 5-year OS at 79% (95% CI: 74.9–83.1) for very young women (<35 years) vs. 83.9% (95% CI: 80.3–87.3) for young women (<40 years) (*p* = 0.145), there was a significantly lower 5-year disease-free survival (DFS) in very young women (53.5%; 95%CI: 48.4–58.6) compared to young women (65.3%; 95% CI: 60.8–69.8; *p* = 0.002) [206]. Compared to women 36–50 years of age, those ≤35 years of age had a higher proportion of HER2 tumours (24.58% vs. 16.94%; *p* = 0.021), PR-(29.85% vs. 22.95%; *p* = 0.043), and stage 3 disease (29.34% vs. 18.52%; *p* < 0.001) [208]. These studies suggest that there may be age-dependent biological differences in BCYW patients, possibly related to changing hormonal and genomic pathways that warrant further research.

Reasons for the poor prognosis of BCYW are likely multifactorial and may be partially explained by the unique tumour biology in younger patients. Younger women are more likely to have tumours of higher grade, larger size, ER/PR negative, and lymph-node positive (*p* < 0.001) [204,205]. Multivariate analysis in a retrospective study including 739 women <40 years of age out of 7105 participants showed that HER2+ tumours (OR = 1.82), nodal involvement (OR = 1.69), histologic grade (grade 3 OR = 1.41), and tumour size (T2 OR = 1.37; T3–T4, 1.47) were independently associated with younger age at diagnosis [208]. The Prospective Study of Outcomes in Sporadic and Hereditary Breast Cancer (POSH) study examined tumour pathology data of 2956 women aged <40, followed for a median of 5 years, and reported that the 5-year-OS in this cohort was 82% with Kaplan–Meier analysis [21]. This study found that ER+ tumours were associated with a significantly higher 5-year OS (5-year OS = 85.0%; 95%CI: 83.2–86.7%) compared to ER- tumours (5-year OS = 75.7%; 95%CI: 72.8–78.4%; *p* < 0.001) [21]. However, 8-year OS was similar in both ER- and ER+ tumours (67.5% vs. 67.7%, *p* = 0.931). In ER+ patients, a flexible parametric survival model for OS showed that the risk of death increases linearly over time, while ER- tumours had the highest risk of death at two years after a breast cancer diagnosis. In a multivariate analysis of 1228 women diagnosed with breast cancer at <40 years of age, TNBC and HER2+ subgroups had significantly poorer overall survival (*p* = 0.0035) [206]. The same study reported a more favourable prognosis associated with negative lymph node involvement (5-year OS HR = 0.68; 95% CI:0.47–0.98; *p* = 0.039) compared to positive lymph node involvement [206]. Even after adjusting for nodal status and breast cancer subtype, younger age is still associated with a lower 5-year disease-free survival (DFS), breast cancer-specific survival, and overall survival compared to older women [5]. Other factors, including the absence of routine screening leading to more advanced stages at presentation and unique genomic mutational profiles, that differ from those of late-onset breast cancer may also contribute to the poor prognosis of BCYW.

Studies have also investigated the prognostic role of *BRCA1/2* pathogenic germline variants in young women [122,209,210,211,212,213,214,215,216,217,218,219,220,221,222,223,224]. Current evidence suggests that carriers of *BRCA1/2* pathogenic variants have similar clinical outcomes as sporadic breast cancer [209]. Similarly, there have been no differences in the survival outcomes reported for breast cancer with or without germline *BRCA1/2* pathogenic variants in young women, except for a trend towards improved survival for carriers of *BRCA1/2* mutations with TNBC compared with non-carriers [122]. The POSH study reported that patients with pathogenic *BRCA1/2* mutations had a similar prognosis as non-carriers. In 2733 women included in the analysis, 2-year OS for *BRCA1/2* mutation carriers was 97.0% (95%CI: 94.5–98.4) versus 96.6% (95%CI: 95.8–97.3) for non-carriers (HR = 0.96; 95% CI: 0.76–1.22; *p* = 0.76) [122]. Similar findings were reported for 5- and 10-year overall survival rates. In 558 patients with TNBC, the *BRCA1/2* mutation carriers (2-year OS = 95%; 95% CI 89–98) had better overall survival than non-carriers (2-year OS = 91%; 95%CI: 88–94) at 2 years (HR 0.59; 95% CI: 0.35–0.99; *p* = 0.047), but no significant differences remained by 5 years or 10 years after diagnosis [122].

Another large international retrospective cohort study that included 1236 breast cancer patients with germline BRCA1/2 mutations diagnosed at age ≤40 years examined clinical outcomes associated with each BRCA mutation type and hormone receptor (HR) status [224]. In this study, the 8-year DFS for *BRCA1* carriers was lower than for the *BRCA2* carriers (62.8% vs. 65.9%, adjusted HR = 0.76, 95% CI 0.60–0.96). However, 8-year OS for BRCA1 (86.9%) and BRCA2 (87.5%) mutation carriers were the same (adjusted HR = 0.69, 95% CI 0.46–1.04). BRCA1 mutation carriers also experienced significantly more frequent second primary malignancies compared to BRCA2 mutation carriers (breast cancer: 17.0% vs. 12.2%, *p* = 0.009; non-breast cancer: 4.3% vs. 1.9%, *p* = 0.02), while distant recurrences were less frequent (10.4% vs. 15.4%, *p* = 0.02) compared to the BRCA2 cohort [224]. Women with HR+ disease had more frequent distant recurrences (*p* < 0.001) and less frequent second primary malignancies (*p* = 0.005 for breast cancer; *p* = 0.18 for non-breast cancer). The study reported no differences in DFS and OS based on HR status [224].

### 3.3. Risk of Local Recurrence and Metastasis

Current evidence suggests that young age is associated with an increased risk for local recurrence (LR) compared to older women and an estimated 7% increase in the risk of locoregional recurrence for every year decrease in age [18]. There are two types of local recurrence: true recurrence that originates from incomplete surgical removal of tumour cells or precancerous lesions, and new primary tumours (i.e., second primary tumour) that are a different histological type or in another location [225]. Local recurrence may be due to lack of radiotherapy, positive margins, and lymphovascular invasion, while other risk factors, including young age at diagnosis, are debated [18,225,226,227,228]. Although rare, local recurrence is concerning due to the increased incidence of distant metastasis and mortality.

Studies have evaluated the local recurrence rates between different follow-up durations and surgical removal techniques [229,230,231,232,233,234,235,236]. A meta-analysis by He et al. analyzed 14 studies for 5-year local recurrence rates and 8 studies for 10-year local recurrence rates using random-effects models. After adjusting for publication bias, this study found that women <40 years had a significantly higher risk of local recurrence developed within five years of breast-conserving surgery compared to older patients (5-year RR = 2.21, 95% CI: 1.62–3.02 and 10-year RR = 1.47; 95% CI: 0.96–2.27) [18]. A cohort analysis of 3024 patients aged 18–40 years diagnosed with breast cancer reported a significantly higher 5-year LR of 5.33% in patients who underwent breast-conserving surgery (BCS) compared to 2.63% in the mastectomy cohort (HR = 3.39; 95% CI, 2.03–5.66; *p* < 0.001) [230]. For women diagnosed with early-stage breast cancer ≤35 years of age, one study reported a LR of 3.5% after BCS compared to 3.6% after mastectomy at 5 years follow-up, while another study found a LR of 12.4% versus 7.5% after BCS and mastectomy, respectively, at 11 years of follow-up with no significant difference in overall survival (HR = 0.99, 95% CI: 0.79–1.26) between the two groups [231,232]. Another recent retrospective study compared the cumulative LR in breast cancer patients aged ≤40 years who underwent either breast-conserving surgery (n = 428) or mastectomy (n = 117) followed by adjuvant systemic treatment [229]. After a median follow-up of 91 months, the 10-year cumulative incidence of local recurrence was 9.3% (median interval = 36.5 months post-operatively) [229]. BCS trended towards an increased risk for local recurrence (11.1% vs. 4.1% for mastectomy; *p* = 0.078). The estimated 10-year distant metastasis-free survival (DMFS) after developing local recurrence in both groups was 84.7%. Univariate analysis demonstrated no significant association between local recurrence and histologic grade, hormone receptor status, HER2, biological subtypes, or tumour size [229]. Patients who underwent BCS had an approximately 2.5-fold higher risk for local recurrence than those who underwent mastectomy. Although most studies have consistently found a significantly higher risk of local recurrence associated with BCS compared to mastectomy [229,230,231,232,233,235,236], there is some conflicting evidence reporting no significantly increased risk that may be related to advancements in systemic therapy [233,234]. Temporal analysis of women aged ≤40 years demonstrates a trend of decreasing LR between 1988 and 2010 from 9.8% (95% CI: 7.1–12.5) to 3.3% (95% CI: 0.6–6.0, *p* = 0.006) [233], and a retrospective study analyzing 565 women aged ≤40 years reported no significant difference in LR between the BCS and mastectomy groups after adjusting for hormone receptor, tumour size, lymph node status, and HER2 status [234].

Studies have also investigated the association between local recurrence of BCYW and different BC subtypes. A retrospective cohort study that included 1099 women ≤35 years of age reported that HR-/HER2+ tumours had a significantly higher 10-year local recurrence rate compared to other subtypes (HR = 20.4; 95% CI: 11.8–35.4), irrespective of breast-conserving surgery or mastectomy treatment [237]. However, Cox proportional hazard model analysis found that BC subtype was not an independent prognostic factor for local recurrence. Similarly, other studies found no significant difference in LR according to BC subtypes for BCYW [238,239]. A prospective cohort study including 514 Danish women with breast cancer and subtype classification of tumours by ER-, PR-, HER2, and Ki67 found that the BC subtype had no significant prognostic impact on the 20-year locoregional recurrence for both younger patients aged ≤45 years and older patients [240]. This study also concluded no significant prognostic impact of BC subtypes on 20-year locoregional recurrence in both age cohorts.

Younger women with breast cancer also have a higher risk of developing metastasis [4,241,242,243,244,245,246,247,248,249]. Previous randomized controlled trials of adjuvant chemotherapy or chemoendocrine therapy in premenopausal women have estimated 5-year metastatic breast cancer (MBC) incidence at 6% to 12% [241,242]. However, this is likely an underestimate given patients who develop metastatic disease tend to be sicker and lost to follow-up. A more recent retrospective analysis of a predominantly (62%) node-negative population examined 395 women <40 years of age diagnosed with non-metastatic breast cancer [243]. The 5-year cumulative incidence of MBC was 24% (95%CI: 20–29%), compared to 9% (95% CI 9–10%) for women ≥40 years (*p* < 0.0001) [243]. Independent risk factors that were significantly associated with MBC within five years from diagnosis were age <40 years at diagnosis, having lymph node or adjacent tissue involvement at diagnosis (regional disease), low versus high socioeconomic status, and the presence of a non-breast primary cancer. A significantly greater proportion of young women developed brain metastases compared to older women (15% vs. 8%; *p* = 0.02) and trended towards more bone (42% vs. 33%; *p* = 0.05) and liver metastases (28% vs. 20%; *p* = 0.05). Unique oncogenic signaling pathways characterized by reduced hormone sensitivity and higher HER2 and EGFR expression have been hypothesized to explain the higher proportion of visceral sites of metastasis in young women [4,243,246,247,248,249]. However, there was no significant difference in median overall survival for young women with MBC compared to older women (18 vs. 14 months, respectively; *p* = 0.21) [243]. De Bock et al. conducted a pooled analysis of 36,000 women, with 406 patients aged <40 years, and multivariate Cox regression showed that age <40 years was a significant prognostic risk factor associated with a 79% increased relative risk (HR = 1.79; 95-CI: 1.28–2.51) for metastasis [244]. Another large retrospective study that examined a predominantly (67%) node-positive disease included 652 women aged <35 years at breast cancer diagnosis and found a 10-year distant recurrence rate of 40% [245]. Thus, although young age at diagnosis of non-metastatic BC before age 40 is associated with a higher risk of developing MBC than older women, especially with brain metastasis, survival rates for metastatic disease between young and older women are similar.

**Figure 2 cancers-15-01917-f002:**
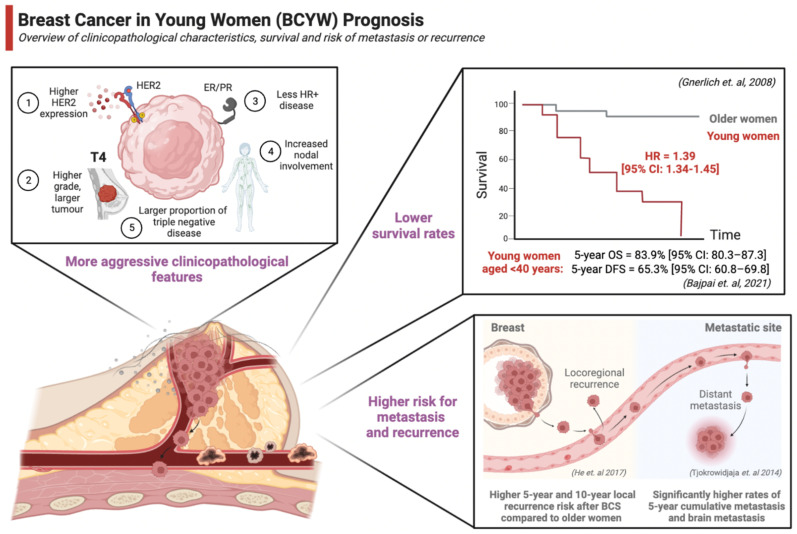
Major differences of breast cancer in young women compared to older women, including more aggressive clinicopathological features, lower survival rates, and increased risk of metastasis and mortality. References: [204,206,225,243].

## 4. Treatment

Multimodal treatment is recommended for breast cancer management in both older and younger women. Many treatment aspects in young women currently lack evidence-based guidelines due to limited age-specific data. Treatment of both early and advanced disease in young women should be based on the same clinicopathological considerations as for older women, while considering the unique biological, hormonal, and genetic characteristics of BCYW [6]. In addition to traditional surgical, radiotherapy, and systemic treatment options, individual treatment planning for young women may also consist of personalized psychosocial support, genetic counselling, fertility, sexual health, and socio-economic consequences (Table 3) [250,251]. Lifelong follow-up care is becoming increasingly relevant for young survivors, given the improved long-term survival with modern therapies [250,251,252].

### 4.1. Locoregional Treatment

#### 4.1.1. Surgical Management

BCYW have similar surgical options as older women: breast conservation therapy (BCT) consisting of partial mastectomy and radiation, or mastectomy. Current data suggest similar survival outcomes for BCT compared to mastectomy, and no survival advantages with contralateral prophylactic mastectomy [296,297,298,299]. Young age alone is not a contraindication to BCT, although the rate of bilateral mastectomies has increased in women of all ages [298,299,300]. This is likely due to improved reconstruction techniques and patient preferences, with recent data suggesting the oncologic safety of immediate and immediate-delayed reconstruction and nipple sparing procedures [301,302]. Other reasons for mastectomy include minimizing the risk of local recurrence after BCS, especially in carriers of pathogenic variants and in locally advanced breast cancer cases [206,300,301].

BCS remains controversial for women younger than 40, given that young age is an independent risk factor for local recurrence after breast-conserving surgery. Previously, mastectomy was the standard of treatment until several randomized control trials reported an equivalent overall survival in patients undergoing BCS with early-stage breast cancer [303,304,305,306,307,308]. A meta-analysis by Vila et al. that included a total of 22,598 patients aged ≤40 years followed for 6–10 years found no significant difference in risk of death after adjusting for nodal status and tumour size for those who underwent BCS compared to mastectomy (HR = 0.90; 95%CI: 0.81–1.00) [253]. Similarly, a SEER database analysis of 7665 women aged <40 years when diagnosed with stage I or II breast cancer with a median follow-up duration of 111 months found no significant differences in 10-year breast cancer-specific survival (BCSS, 87.7%; 95%CI: 86.5–88.9% for BCS vs. 85.4%; 95%CI: 83.8–87.9% for mastectomy) and 10-year overall survival (85.9%; 95% CI: 84.5–87.3% for BCS and 83.5%; 95%CI: 81.9–85.1% for mastectomy) [254]. However, the 35–39 years age group was associated with higher 10-year BCSS (88%) and OS (86.1%) compared with the younger patients aged 20–34 years (10-year BCSS = 84.1% and 10-year OS = 82.3%; *p* < 0.001 for both BCSS and OS). Importantly, the BCS group demonstrated noninferior 10-year BCSS and OS compared with mastectomy in a subgroup analysis according to stage of disease for both 35–39 years and 20–34 years age groups [254]. In a retrospective analysis of 1520 patients aged ≤35 years with a median follow-up of 5.1 years, multivariate Cox analysis showed that BCS was associated with a significantly improved 5-year DFS (HR = 0.45; 95% CI: 0.28–0.73; *p* = 0.001) and OS (0.41; 95% CI 0.21–0.80, *p* = 0.009) compared to mastectomy. This study also reported an improved 5-year DFS and OS for patients with mastectomy and reconstruction compared to mastectomy alone; however, this difference was not significant [255].

#### 4.1.2. Radiation Therapy

Adjuvant whole breast radiation after partial mastectomy is associated with a lower risk of ipsilateral breast tumour recurrence and improved breast cancer survival [258,304]. Adjuvant radiation is important for young women who may derive greater clinical benefit due to their higher absolute risk of local recurrence compared to older women [258]. Care should be taken to minimize radiation exposure to adjacent organs and reduce the risk of late adverse effects and secondary malignancies whenever possible, given their young age and long-term survival potential.

Based on the current literature, the indications and schedules for hypofractionation in younger women should be the same as in other age groups [309,310]. However, there are important considerations for treatment volume, as accelerated partial breast irradiation (APBI) has insufficient evidence supporting its application to younger women. Several guidelines include a recommended minimum patient age of 45–60 years in the APBI selection criteria [311,312,313], based on existing published studies primarily including women >50 years. There are concerns for increased risk of multifocal and multicentric disease and higher local recurrence rates in younger women that may decrease the effectiveness of APBI in this population. The NSABP B-39 trial aims to compare whole and partial breast radiation in women aged >18 years; however, the results have yet to be published [314].

Currently, the standard of care for young women comprises whole-breast radiation with standard fractionation; although, the use of boost may also be beneficial. The European Organization for Research and Treatment of Cancer trial found that using boost was associated with reduced 20-year ipsilateral breast tumour recurrence from 16.4% to 12.0% (HR = 0.65, *p* < 0.0001) [256]. This study reported the greatest benefit of boost in young women <40 years of age at 11.6%, compared to 4.4% in the general cohort. Although hypofractionated whole-breast schedules have been shown to be as effective as standard fractionation, most of the studies only included women aged >50 years, which is currently the recommended age for hypofractionation [315].

For young women with locally advanced breast cancer, adjuvant radiation following mastectomy is associated with improved local control and overall survival. A retrospective analysis of 107 women aged ≤35 years with stage II or III breast cancer showed that patients who received postmastectomy radiotherapy (PMRT) had better 5-year locoregional control (88% vs. 63%, *p* = 0.001) and 5-year overall survival (67% vs. 48%, *p* = 0.03) compared with those who did not [257]. Since young age is a risk factor for locoregional recurrence after mastectomy, studies suggest that young women with node-negative disease and additional risk factors may still benefit from postmastectomy radiation [316,317,318]. Similarly, several clinical trials have demonstrated a locoregional recurrence and survival benefit in both premenopausal and postmenopausal women receiving adjuvant radiation with large primary tumours >5 cm, invasion of the skin or chest wall, or lymph node involvement [319,320,321].

### 4.2. Systemic Treatment

#### 4.2.1. Endocrine Treatment

Approximately 60–75% of breast cancer cases in young women are hormone receptor (HR) positive with either ER and/or PR receptor positivity [322]. In addition to surgery and chemotherapy, the standard of care for adjuvant therapy in young women who have HR+ breast cancer is tamoxifen. Tamoxifen is a selective ER-alpha modulator (SERM) that blocks ER signalling and interferes with cancer cell proliferation. Aromatase inhibitors (AI) are another endocrine therapy commonly used in postmenopausal but not premenopausal women due to the limited ability to reduce circulating estrogen. Unlike postmenopausal women, premenopausal women have higher amounts of aromatase substrate in the ovaries. AI administration with decreased peripheral estrogen production leads to positive feedback stimulating gonadotropin release, resulting in increased ovarian estrogen production [323]. Hence, AIs are considered to be ineffective for premenopausal women without ovarian suppression.

Neoadjuvant endocrine therapy is currently not routinely recommended for young women [6,290,324,325]. In the adjuvant setting, Early Breast Cancer Trialists’ Collaborative Group’s (EBCTCG) meta-analysis of 194 clinical trials examining adjuvant chemotherapy or endocrine therapy in women of all ages with breast cancer reported that five years of adjuvant tamoxifen was associated with a lower annual breast cancer death rate by approximately one-third, and recurrence risk (including local and distant) by 50% for women with HR+ cancers irrespective of age [258]. However, given that approximately 10% of patients with HR+ disease subsequently develop relapse after five years, several studies have compared 5-year versus 10-year duration of tamoxifen therapy [259,260]. The aTTom and ATLAS trials suggest a modest but delayed benefit in recurrence-free and overall survival for women on extended tamoxifen therapy. Both studies reported a time-dependent relapse risk reduction, with almost no benefit from longer treatment during years 5–9 of adjuvant tamoxifen treatment, followed by a significant improvement in years ten and beyond [259,260]. Specifically, the recurrence rate ratio (RR) in the ATLAS trial was 0.90 (95% CI: 0.79–1.02) during years 5–9 and 0.75 (95% CI: 0.62–0.90) ≥10 years; while, breast cancer mortality RR was 0.97 (95% CI: 0.79–1.18) during years 5–9 and 0.71 (95%CI 0.58–0.88) in later years for women treated with extended tamoxifen compared to those who were not [260]. Similarly, aTTom reported a RR of 0.99 during years 5–9 (95% CI: 0.86–1.15) and 0.75 (95%CI: 0.66–0.86) at ≥10 years [259]. A pooled analysis of aTTom and ATLAS demonstrated a 3% lower risk of death for patients who completed ten versus five years of adjuvant tamoxifen during years 5–9 (RR = 0.97; 95%CI: 0.84–1.15) and an increased relative risk reduction to 25% starting at year 10 (RR = 0.75; 95%CI: 0.65–0.86) [326].

Recent clinical trials have studied the optimal adjuvant endocrine therapy including ovarian suppression in young women with HR+/HER2- breast cancer. The ABCSG-12 Trial, which randomized 1803 premenopausal women with stage I or II ER/PR+ breast cancer to adjuvant anastrozole or tamoxifen with a median follow-up of 47.8 months, reported a disease-free survival (DFS) rate of 92.8% in the tamoxifen group and 92.0% in the anastrozole group [261]. There was no significant difference in the 7-year disease-free survival and overall survival for the anastrozole compared to tamoxifen groups (HR = 1.10; 95%CI: 0.78 to 1.53; *p* = 0.59 and HR = 1.80; 95% CI: 0.95–3.38; *p* = 0.08, respectively) [261]. However, the combination of ovarian function suppression (OFS) with triptorelin plus tamoxifen or AI was shown to improve outcomes. The parallel SOFT and TEXT trials combined and randomized a total of 4690 premenopausal women with ER/PR+ breast cancer receiving OFS to adjuvant AI (exemestane) or tamoxifen [262,263]. Unlike the ABCSG 12 trial, the combined results found a higher 8-year disease-free survival in the exemestane + OFS (86.8%) compared to tamoxifen + OFS group (82.8%) (HR for recurrence, a second invasive cancer, or death = 0.77; 95% CI: 0.67–0.90; *p* < 0.001) [262,263]. The 8-year overall survival was not significantly different between the two groups (93.4% for exemestane + OFS vs. 93.3% for tamoxifen + OFS) with a hazard ratio for death of 0.98 (95% CI, 0.79 to 1.22; *p* = 0.84) [262,263]. Interestingly, the SOFT trial also found that for women who received chemotherapy and remained pre-menopausal, adding OFS to tamoxifen was associated with an absolute increase in 5-year breast cancer-free survival of 4.5% (78% in tamoxifen alone vs. 82.5% in tamoxifen + OFS) [262]. The benefit was greater for the exemestane + OFS group compared to the tamoxifen alone group, with an increased absolute 5-year breast cancer-free survival and 5-year distant recurrence-free survival of 7.7% and 4.2%, respectively [262]. Subgroup analysis demonstrated similar treatment effects regardless of whether patients received chemotherapy, but the largest absolute benefit for OFS over tamoxifen alone occurred in younger women who remained premenopausal after previous chemotherapy [262]. Among women aged <35 years, 5-year breast cancer-free survival in the tamoxifen alone arm was 67.7% vs. 78.9% in tamoxifen + OFS and 83.4% in exemestane + OFS; although, there has been no significant improvement in overall survival. Similarly, a recent meta-analysis of four trials (ABCSG XII, SOFT, TEXT, and HOBOE trials) aimed to investigate the benefit of aromatase inhibitors for premenopausal women on ovarian suppression given their superior effectiveness compared to tamoxifen in the postmenopausal setting [275]. Among 7030 premenopausal women with ER+ tumours receiving OFS, there was a lower rate of breast cancer recurrence for women who received AI and ovarian suppression compared to those who received tamoxifen (RR = 0.79; 95% CI 0.69–0.90, *p* = 0.0005), with the highest benefit in years 0–4 and no significant difference between groups observed beyond five years [264]. AI with OFS may be a reasonable treatment option for younger women that reduces breast cancer recurrence; although, longer follow-up is needed to assess the impact on breast cancer mortality.

#### 4.2.2. Chemotherapy

The indication and regimens for chemotherapy and HER2-targeted therapies in younger women are the same as in older women. Therefore, treatment decisions are guided by disease stage, biological features, and patient preferences, as young age alone is not an indicator for more aggressive treatment.

Indications for neoadjuvant chemotherapy (NAC) in BCYW are the same as for the general population: inoperable HR+ disease, or preoperative downstaging of locally advanced breast tumour, or axillary nodal disease. Evidence suggests that young women benefit more from NAC than older women. The GeparTrio trial randomized 2072 women with operable or locally advanced breast cancer to a course of neoadjuvant TAC chemotherapy regimen based on their early clinical response. This trial reported the highest pathological complete response (pCR) rate—defined as no residual invasive tumour in the breast and no involved lymph nodes at surgery—in women aged <40 years with TNBC or grade 3 tumours compared to women aged ≥40 years (57.0% vs. 34.0 % respectively; *p* < 0.0001) [265]. When compared to women aged ≥40 years, women <40 years had approximately double the odds of pCR at surgery (OR = 2.02; 95% CI: 1.569–2.610; *p* < 0.0001) [265]. Similarly, a pooled analysis of eight randomized controlled trials that included a subgroup of 1453 women aged <40 years compared to those aged 40–49 and ≥50 years found a significantly higher pCR in the young (<40 years) group compared with older groups (20.9 vs. 17.7 vs. 13.7%, respectively; *p* < 0.001). Specifically, young women with HR+/HER2- and TNBC disease were more likely to achieve pCR after NAC compared to older women (11% for <40 years vs. 5.8% for ≥50 years; *p* < 0.001 in HR+/HER2- and 39.3% for <40 years vs. 25.2% for ≥50 years; *p* < 0.001 in TNBC) [266].

Although women aged <40 years had better pCR, indicating chemotherapy response, older women (≥50 years) had significantly better survival outcomes with improved OS (HR = 0.87; 95% CI: 0.74–1.02; *p* = 0.079), LRFS (HR = 0.64; 95%CI: 0.52–0.79; *p* < 0.001), and DFS (HR = 0.81; 95%CI: 0.71–0.92; *p* = 0.001) compared to young women <40 years [266]. More aggressive tumour biology and worse survival for young women who do not attain pCR with HR + /HER- and TNBC and lower efficacy of adjuvant therapy (i.e., endocrine therapy for HR+ disease) may explain the worse recurrence and survival rates despite higher pCR rates in young women. For example, a Korean population-based study analyzing 1444 women aged <35 years with breast cancer found that young women were less likely to benefit from adjuvant hormone therapy with higher de novo tamoxifen resistance compared to older women [19]. Although results from other trials have demonstrated that endocrine therapy in young women is effective [323,327], these studies focused primarily on the adjuvant setting with women receiving OFS, which is not routinely recommended or provided in studies on NAC. Another study that included 170 young women aged ≤40 years at diagnosis who received NAC for stage II–III BC between 1998 and 2014 reported that attaining pCR was associated with significantly improved 5-year DFS at 91% versus 60% for those without pCR (HR = 0.12; 95% CI: 0.04–0.39; *p* < 0.001), irrespective of receptor status [267]. Achievement of pCR was also associated with significantly improved OS (HR = 0.19; 95% CI: 0.06–0.62; *p* = 0.006), with a 5-year OS for patients with pCR of 95% versus 75% for those without pCR for all receptor subtypes [267]. Given the potential toxicities associated with NAC in young women with long life expectancies, de-escalation of therapy after attaining pCR and salvage therapy when pCR is not achieved are important considerations. Recent trials (CREATE-X, NCT02445391 and PENELOPE-B) are investigating capecitabine, platinum agents, and targeted therapies including CDK4/6 inhibitors in patients without pCR following NAC; however, further research is required for young women [268,269,270].

The benefit of adjuvant chemotherapy in young women with low-risk, early-stage HR+ breast cancer on optimal ET is unclear. Gene expression signatures, such as Oncotype Dx, MammaPrint, Prosigna, Endopredict, and Breast Cancer Index, can guide clinical decision-making by predicting patients’ individual recurrence risk and benefit from chemotherapy in postmenopausal BC patients. Guidelines state that young women with low-risk gene expression scores and favourable clinicopathological features tend to have excellent survival outcomes and may consider omitting chemotherapy [328]. However, according to TAILORx and MINDACT trials in node-negative patients and the RxPONDER trial in node-positive patients, younger age women with low-intermediate gene expression profiling scores still benefit from chemotherapy [271,272,273]. TAILORx evaluated the 21-gene Oncotype Dx recurrence score (RS) amongst 10,273 women with HR+, HER2-, T1-2, and node-negative disease and categorized women as low, intermediate, or high risk of recurrence [271]. The 5-year distant recurrence-free survival (DRFS) was 98.0% for those with low-risk RS on ET alone compared to 98.2% for those with low-risk RS on ET and chemotherapy. The 9-year DRFS was 94.5% for the ET-only group and 95.0% for the chemotherapy + ET group in those with intermediate-risk RS, indicating no benefit for adding chemotherapy to patients with low-intermediate RS [271]. Exploratory subgroup analyses of women aged ≤50 years found a benefit (percentage difference in distant recurrence was 0.8–3.2% at five years and 1.6–6.5% at nine years) from chemotherapy amongst those with an intermediate RS of 16–25, although overall survival rates were similar [271]. Similarly, the RxPONDER trial randomized 5015 (3350 postmenopausal and 1665 premenopausal) women with stage II/III breast cancer, one to three involved nodes, and a RS ≤ 25 to either endocrine therapy or endocrine therapy plus standard chemotherapy. In contrast to postmenopausal women who did not benefit from adjuvant chemotherapy, premenopausal women who received chemoendocrine therapy had longer 5-year invasive disease–free survival (93.9% in the chemoendocrine group vs. 89.0% in the endocrine-only group; absolute difference = 4.9%) and distant relapse–free survival (HR = 0.58; 95% CI, 0.39 to 0.87; *p* = 0.009) than those who received endocrine therapy alone. There was a significant chemotherapy benefit for premenopausal women with a hazard ratio for invasive disease recurrence, new primary (breast or other) cancer or death of 0.60 (95% CI: 0.43–0.83, *p* = 0.002) in the chemoendocrine group compared to the endocrine-only group [272]. The MINDACT trial evaluated the 70-gene signature that classifies women into low or high risk for recurrence, irrespective of HR status, based on both clinical and genomic risk [273]. This study enrolled 6693 women aged 18–70 years with localized breast cancer (stage T1, T2, or operable T3) with up to three positive lymph nodes and randomized those with high clinical risk and low genomic risk to receive chemotherapy or not. This trial reported 8-year distant metastasis-free survival rates classified as high risk of 92.0% (95%: CI 89.6–93.8) for the chemotherapy cohort versus 89.4% (95%CI: 86.8–91.5) for no chemotherapy (HR = 0.66; 95% CI:0. 48–0.92) [273]. An exploratory analysis in 1358 patients, classified as low risk with HR+ and HER2- disease, found different effects of chemotherapy administration according to age: 8-year distant metastasis-free survival of 93.6% (95% CI: 89.3–96.3) with chemotherapy versus 88.6% (95%CI: 83.5–92.3) without chemotherapy in women aged ≤50 years (absolute difference = 5.0%), and 90.2% (95%CI: 86.8–92.7) versus 90.0% (95%CI: 86.6–92.6) in women >50 years (absolute difference = 0.2%) [273]. Although these results suggest that the magnitude of the benefit from adding adjuvant chemotherapy to ET may be age-dependent, reaching a clinically relevant threshold of 5% in women aged ≤50 years, this was an underpowered exploratory analysis and further studies are necessary to confirm these findings. To date, there are no commercially available and validated prognostic genomic assays for HR+ early breast cancer, and further research is necessary to develop assays to predict the most appropriate type of treatment based on genomic risk.

Young women with high-risk breast cancer are treated with the same combination regimens, including anthracyclines and taxanes used for older women. The Preferred National Comprehensive Cancer Network (NCCN) guidelines for adjuvant chemotherapy for high-risk HR- breast cancer patients recommend regimens with doxorubicin plus cyclophosphamide followed by paclitaxel or docetaxel plus cyclophosphamide [329]. The EBCTCG meta-analysis evaluated the benefits of adjuvant polychemotherapy on outcomes in women aged <50 years versus 50–69 years [264]. This study found that anthracycline-based chemotherapy was associated with a larger reduction in the annual breast cancer death rate in younger compared to older women (38% vs. 20%, respectively), independent of hormone-receptor status, tamoxifen use, nodal status, and other tumour features [264]. Results from another meta-analysis of 13 studies including 22,903 patients found that adding taxanes to anthracycline-based chemotherapy was associated with improved disease-free survival (pooled HR = 0.83; 95% CI: 0.79–0.87; *p* < 0.00001) and overall survival (pooled HR = 0.85; 95% CI: 0.79–0.91; *p* < 0.00001) in both younger and older women with high-risk early stage breast cancer [274]. A large retrospective study including over 10,000 women aged <50 years reported no age-specific differences in survival among women who received chemotherapy. However, younger women who did not receive chemotherapy aged <40 years had a higher relative risk of death at 10-year follow-up compared to those aged 45–49 years (RR = 1.40; 95%CI: 1.10–1.78 for ages 35–39 and RR = 2.18; 95% CI: 1.64–2.89 for age <35) [15]. There are data to suggest that select populations of young women can achieve excellent clinical outcomes with endocrine therapy alone; thus, young age alone should not be a sole indicator for chemotherapy [330,331,332]. However, we need better tools for differentiating those patients who can spare chemotherapy among young women with breast cancer.

#### 4.2.3. Targeted Therapy

Younger women with TNBC and HER2+ breast cancer are also treated according to general guidelines for older breast cancer patients. For TNBC, the standard neoadjuvant chemotherapy regimen consists of adriamycin, cyclophosphamide, and paclitaxel that has a pathologic complete response (pCR) of 35–45% [327]. For TNBC, recent trials have suggested improved clinical outcomes with programmed death-ligand 1 (PD-L1) inhibitors in combination with standard chemotherapy that were consistent across subgroups, including PD-L1 expression, tumour size, and nodal involvement subgroups [275,276,277]. For HER2+ breast cancer, one-year treatment with adjuvant trastuzumab combined with standard chemotherapy is indicated irrespective of age for all node-positive or high-risk node-negative breast cancers (tumour size > 0.5 cm) and no significant cardiovascular contraindications [328].

In young patients with TNBC, the landmark KEYNOTE trial demonstrated that adding pembrolizumab to neoadjuvant anthracycline–taxanes–platinum chemotherapy is associated with a significantly higher percentage of pCR compared to neoadjuvant chemotherapy only (64.8% vs. 51.2%; *p* < 0.01) [275]. Importantly, this benefit was seen irrespective of the PD-L1 expression levels. Similarly, the phase II GeparNuevo trial evaluated whether adding the PD-L1 inhibitor durvalumab to neoadjuvant chemotherapy influenced complete response or survival in early-stage TNBC patients without previous treatment. This study randomized 117 participants (27% aged <40 years) and reported a modest improvement in pCR in the durvalumab group vs. chemotherapy only (61% vs. 41%; OR = 2.22; 95% CI: 1.06–4.64; *p* = 0.035). There was no significant increase in pCR rates in PD-L1+ tumours compared to PD-L1-ones [276]. Durvalumab combined with neoadjuvant chemotherapy was associated with significantly improved DFS (91.4% vs. 79.5%; HR = 0.37; 95%CI: 0.15–0.87; *p* = 0.0148) [276]. The NeoTRIPaPDL1 trial found no significant difference in the pCR (42.3% for placebo vs. 47.1% for atezolizumab group; *p* = 0.66) in 279 patients with early-stage TNBC and no specific age group was randomized to receive neoadjuvant chemotherapy alone versus with atezolizumab [277]. Future studies are needed to study the effects in young women. There was a trend towards improved pCR with higher PD-L1 expression levels (55% for ≥5% PD-L1 vs. 32% for 1–5% PD-L1; *p* = 0.148) [277]. Although emerging evidence suggests that adding an immune checkpoint inhibitor increases pCR, regardless of PD-L1 activity, future studies are required to evaluate this effect in young women.

The standard HER2-targeted agents are the same for young and older women, including trastuzumab alone, trastuzumab with pertuzumab, or emtansine in certain patients. In patients with HER2+ tumours, the addition of trastuzumab to chemotherapy has reduced recurrence rates and breast cancer-related mortality by approximately a third, irrespective of patient age and tumour characteristics [278]. Among the studies that have evaluated shorter duration regimens of trastuzumab, only one study reported noninferiority of 6 months versus one year of trastuzumab treatment [279]. A meta-analysis evaluating shorter duration of trastuzumab compared with one year of treatment concluded that one year of treatment was superior [280]. Although one year of trastuzumab treatment remains the standard of care, another recent meta-analysis concluded that a shorter duration of adjuvant trastuzumab was associated with noninferior 5-year disease-free survival to one year administration and resulted in lower rates of cardiotoxic side effects. Given the conflicting evidence, current guidelines state that in certain low-risk patients, shorter trastuzumab duration can be discussed on an individual basis [328].

Double HER2-blockade therapy with pertuzumab and trastuzumab for adjuvant may be considered for HER2+ patients at high risk of relapse, but there are currently no data on its efficacy in young women specifically or in the neoadjuvant setting [328]. In the APHINITY study (13.6% of patients aged <40 years in each treatment arm), adjuvant pertuzumab in addition to trastuzumab and standard chemotherapy was associated with a significant improvement in the 3-year rate of invasive-disease-free survival compared to trastuzumab and chemotherapy alone (92.0% vs. 90.2%; HR = 0.77; 95% CI: 0.62–0.96; *p* = 0.02) among patients with HER2+, node-positive breast cancer [281].

For HER2+ young breast cancer patients with residual pathological disease after completing neoadjuvant chemotherapy and anti-HER2 therapy, the standard of care is currently one year of adjuvant anti-HER2 therapy with trastuzumab–emtansine (T-DM1) [333]. T-DM1 is an antibody–drug conjugate of trastuzumab and the cytotoxic agent emtansine (DM1) that retains trastuzumab activity while delivering DM1 intracellularly to HER2+ cells. Interim analysis of the KATHERINE study (20% of patients aged <40 years in each treatment arm) showed that patients who received adjuvant T-DM1 had a significantly improved 3-year invasive disease-free survival of 11.3% compared to those who received standard trastuzumab (invasive disease or death HR of 0.50; 95% CI: 0.39–0.64; *p* < 0.001), irrespective of the extent of residual disease, HR status, and type of neoadjuvant HER2-targeted therapy [282].

#### 4.2.4. Chemotherapy for Loco-Regional Relapse and Metastasis

Given that young age is a risk factor for local relapse, margin status should be carefully monitored in young women [334]. The results of the CALOR trial demonstrated a significantly improved 5-year disease-free survival in women who received chemotherapy for isolated locoregional recurrence (ILRR) after surgical removal with negative margins, and 5-year DFS of 69% (95% CI 56–79) with chemotherapy versus 57% (95%CI: 44–67) without chemotherapy (HR = 0.59; 95% CI: 0.35–0.99; *p* = 0.046) [283,284]. Specifically, chemotherapy was significantly more effective in women with resected ER- tumours. ET is recommended for locally recurrent ER+ disease and trastuzumab for HER2+ disease for younger and older women; although, this is based on expert opinion and requires higher quality evidence [328].

HR positive disease represents 65.9% of MBC cases in women aged <40 years [21]. First-line endocrine therapy for premenopausal MBC is tamoxifen alone or an LHRH agonist alone; although, combination therapy may provide clinical benefit. In a meta-analysis of four studies comparing LHRH agonist with or without tamoxifen in premenopausal women with metastatic breast cancer, there was a statistically significant improved median survival (2.9 years with combination vs. 2.5 years with LHRH agonist alone; *p* = 0.02) and median PFS with combined therapy versus monotherapy (8.7 vs. 5.4 months, *p* = 0.0003) [285]. Similar comparisons of tamoxifen alone versus combination with LHRH agonist have not reported the same findings; however, adding tamoxifen should be considered in patients receiving LHRH agonist given these results. AIs are indicated for young women in the first-line metastatic setting only if they are ovarian suppressed [335]. Several phase II small studies have demonstrated clinical benefit and efficacy for premenopausal women treated with LHRH agonist and aromatase inhibitors [286,287,288,289,290]. Emerging evidence for novel treatments of advanced or metastatic HR+ breast cancer in post-menopausal women, including fulvestrant (SERM), letrozole with palbociclib (CDK 4/6 inhibitor), and everolimus (mTOR inhibitor) may be extrapolated to younger women [336,337,338,339,340,341,342,343]. However, there are no data for these treatments specific to young women, and further studies are warranted.

There is a higher proportion of TNBC and HER2+ disease in young women with metastatic breast cancer [344,345]. The first-line treatment is chemotherapy, with or without targeted therapy. For patients who progress onto anthracycline-based therapy, significantly improved overall survival has been reported for capecitabine/docetaxel and gemcitabine/paclitaxel [346,347]. These regimens may be more appropriate for younger women who are likely healthier and better able to tolerate the associated toxicities. Eribulin, a non-taxane microtubule inhibitor, is also approved for MBC in patients who progress after at least two lines of anthracycline- and taxane-based chemotherapy following the EMBRACE trial [291]. This trial reported a significantly improved overall survival in the eribulin group compared with a treatment of the physician’s choice (median OS of 13 vs. 11 months; HR = 0.81, 95% CI: 0.66–0.99; *p* = 0.041). Other standard chemotherapy regimens following progression on anthracyclines and taxanes include capecitabine, liposomal doxorubicin, ixabepilone, or vinorelbine either alone or in combination. Taxane rechallenge has been shown to have good activity, with a 37–51% response rate [348]. Oral etoposide and metronomic cyclophosphamide and methotrexate are also reasonable options; however, guidelines for the optimal sequencing are lacking [328,349,350].

There are several new advancements in metastatic TNBC treatment. Recently, pembrolizumab received Food and Drug Administration (FDA) approval to be used with chemotherapy for the treatment of locally recurrent unresectable or metastatic TNBC with PD-L1+ expression following the KEYNOTE-355 trial [275]. This trial reported significantly improved progression-free survival in the pembrolizumab plus chemotherapy arm compared to chemotherapy alone, with median PFS of 9.7 months and 5.6 months, respectively (HR = 0.65; 95% CI: 0.49, 0.86; *p* = 0.0012). In addition, poly(adenosine diphosphate-ribose) polymerase inhibitors (PARPi) may be useful in BRCA-mutated breast cancers, which are seen more frequently among TNBC cases. Olaparib and talazoparib are PARPi monotherapies approved for patients with deleterious germline BRCA-mutated, HER2-metastatic breast cancer. The OlympiAD study demonstrated a superior response rate and PFS of olaparib with a more favourable toxicity profile and OS benefit of 7.9 months (22.6 versus 14.7) for patients who did not receive chemotherapy in the metastatic setting [292]. The EMBRACA trial had a similar design and demonstrated the superiority of talazoparib with a significantly improved PFS (8.6 months vs. 5.6 months; HR = 0.54; 95%CI: 0.41–0.71; *p* < 0.001) compared to standard single-agent therapy [293]. Other novel agents including enzalutamide, which interferes with androgen receptor signaling, and the antibody drug conjugate sacituzumab govitecan have demonstrated promising preliminary data; however, future clinical trials are required [351,352].

In patients with HER2+ metastatic breast cancer, pertuzumab added to trastuzumab/docetaxel has been shown to significantly prolong both progression-free survival and overall survival. In the CLEOPATRA trial, the median overall survival was 56.5 months with an improved overall survival by 15.7 months in the pertuzumab added group compared to those who received only trastuzumab/docetaxel [294]. Hence, adding pertuzumab has become the standard of care, with younger patients being more likely to tolerate this rigorous regimen [294,353]. HER2-targeting agents are also recommended after progression on trastuzumab [354,355,356]. T-DM1 is approved for use in patients with HER2+ metastatic breast cancer who previously received treatment with trastuzumab and a taxane. In the EMILIA trial comparing TDM-1 to the combination of capecitabine and lapatinib, TDM-1 showed an OS benefit of 30.9 versus 25.1 months [295]. These therapies have greatly improved the treatment of metastatic HER2+ disease. Previously, HER2+ disease had a very poor prognosis due to its aggressive nature; however, the advent of newer HER2 targeting agents has led to improved overall survival for up to 4–5 years [357].

## 5. Special Considerations

### 5.1. Fertility

Unique considerations for BCYW include treatment-induced ovarian failure and infertility. Infertility risk varies according to age, reproductive reserve, and aspects of treatment including chemotherapy agent, duration of treatment, and dose administered [358,359]. Accurate estimates of infertility after breast cancer treatments are difficult to ascertain, with studies using different surrogates including amenorrhea, estradiol, anti-Mullerian hormone, inhibin B, and follicle count [360,361].

There is evidence supporting the efficacy and safety of temporary OFS with GnRHa during chemotherapy to preserve ovarian function, with similar disease outcomes [362,363]. The POEMS study randomized 257 premenopausal women with HR- breast cancer to standard chemotherapy with or without goserelin to determine if goserelin reduced ovarian failure [364]. The ovarian failure rate was 8% in the intervention group versus 22% in the chemotherapy group. This study also reported that 22 patients in the goserelin group achieved at least one pregnancy versus 12 in the standard group. Although limited to HR- breast cancer, these promising results have led to recommendations that concomitant GnRHa with neoadjuvant or adjuvant chemotherapy should be offered to all patients who wish to preserve ovarian function. However, there is still insufficient evidence for the efficacy of GnRHa for fertility protection; therefore, current guidelines state that GnRHa use during chemotherapy cannot replace fertility preservation methods, which should be offered to all young patients [328]. These include cryopreservation of embryo, oocyte, or ovarian tissue.

### 5.2. Breast Cancer during Pregnancy

Breast cancer is the most common pregnancy-associated malignancy, with an incidence of pregnancy at the time of breast cancer diagnosis of approximately 1.5% [365,366,367].

Surgery is considered the safest treatment modality during early pregnancy (first and second trimester) as endocrine and cytotoxic therapies are contraindicated during this period. Sentinel lymph node biopsy may be pursued with minimal risk for fetal harm using technetium. Patients who undergo surgery early in pregnancy and do not require adjuvant chemotherapy may receive delayed radiation a few months later, while there might be worse outcomes with such delays [368]. As a result, mastectomy is recommended unless the patient is in their third trimester and is scheduled to deliver before causing delays to radiation therapy. Furthermore, fetal radiation exposure is associated with birth defects, mental retardation, and childhood malignancy, among others. Radiation therapy is contraindicated in the third trimester, as the potential risk for fetal radiation exposure increases with fetal growth closer to the breast [368,369].

Fetal malformations are also highly associated with chemotherapy exposure during the first trimester [370]. Anthracycline-based regimens have been extensively studied and may be given during the second and third trimesters [359,371,372]. However, chemotherapy should ideally be held at least three weeks before delivery to avoid cytopenias. Guidelines recommend avoiding pregnancy within six months of systemic therapy completion given its teratogenicity [373,374]. Other systemic therapies, including endocrine therapy and anti-HER2 therapy, are contraindicated during pregnancy. Attempting pregnancy should be avoided in those with HR+ disease until at least 18–24 months of endocrine therapy have been completed [328]. Retrospective studies have suggested no worsening of breast cancer outcomes in patients who have become pregnant [374,375,376]. POSITIVE (Pregnancy Outcome and Safety of Interrupting Therapy for women with endocrine responsIVE Breast Cancer) is an ongoing trial, aiming to recruit 500 premenopausal women with ER+ early breast cancer aged ≤42 years to evaluate the safety and outcomes of women with HR+ cancers who interrupt endocrine therapy for childbearing purposes [377].

### 5.3. Bone Health

In young women receiving endocrine therapy for breast cancer, bone mineral density changes have been reported with tamoxifen, ovarian suppression, tamoxifen or aromatase inhibitors with ovarian suppression, and oophorectomy [378,379,380]. Although more common in older women, younger women may also experience bone compromise secondary to hormonal, chemotherapy, and radiation treatment for breast cancer. Pharmacologic treatment with bisphosphonates or RANK ligand inhibitors is indicated for patients with a history of osteoporosis, fragility fractures, and osteopenia, with additional risk factors [328]. Several studies have shown antiresorptive therapies to maintain or increase bone mineral density in women treated with endocrine therapy [381,382,383,384,385,386]. Guidelines recommend consideration of adjuvant bisphosphonates in young women receiving OFS [328]. Although there was no major teratogenicity associated with recent bisphosphonate exposure in pregnant women, a recent case-control study found increased rates of neonatal complications and spontaneous abortions [387,388]. Caution should be taken for women interested in future fertility, given the long half-life of bisphosphonates.

## 6. Conclusions and Future Directions

Breast cancer in young women is diagnosed in more advanced stages and has more aggressive biological features. To date, there are limited age-specific data for the epidemiology, clinicopathologic characteristics, outcomes, and treatments of breast cancer in young women. Geographic, ethnic, and racial variations need to be studied to identify differences in breast cancer and treatment in different populations of young women. Further, data on the effects of physical activity, socioeconomic status, and other lifestyle-related factors (drinking, smoking, and BMI) are required to guide future clinical care of young patients. Existing treatment guidelines for young women are derived from studies conducted among older populations who have distinct tumour biology that is associated with a better prognosis. Using advanced molecular genetic technologies to identify key driver mutations for selecting appropriate targeted therapy and monitoring response to treatment can provide more personalized treatment for young women. Further age-specific studies such as Reducing the Burden of Breast Cancer in Young Women (RUBY) are warranted to help decrease the burden and improve clinical outcomes for breast cancer in young women.

## Figures and Tables

**Figure 1 cancers-15-01917-f001:**
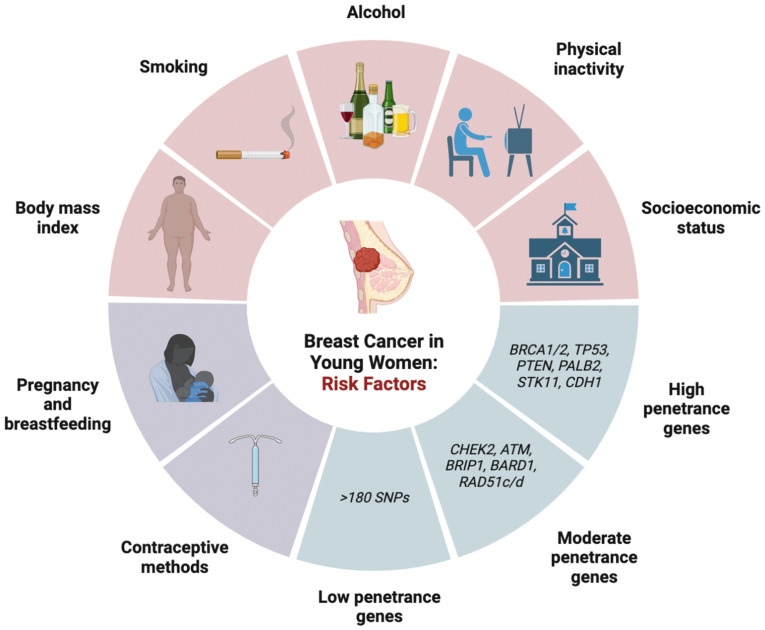
Risk factors associated with breast cancer in young women of onset development including lifestyle risk, genetic risk factors, and reproductive risk factors. SNPs: single nucleotide polymorphisms.

**Table 1 cancers-15-01917-t001:** Breast cancer susceptibility genes and breast cancer risk in young women from recent major population-based studies.

Breast Cancer Susceptibility Gene	Study, Year	Patient Population (Age Group)	Odds Ratio	95% Confidence Interval	Penetrance
BRCA1	Breast Cancer Association Consortium, 2021 [118]	175 cases, 10 controls with mutation (<40 years)	32.8	16.9–63.4	High
	Hu et al., 2021 [119]	209 cases, 27 controls with mutation (<45 years)	8.63	5.63–13.89	High
BRCA2	Breast Cancer Association Consortium, 2021 [118]	156 cases, 20 controls with mutation (<40 years)	11.9	7.33–19.4	High
	Hu et al., 2021 [119]	296 cases, 38 controls with mutation (<45 years)	7.65	5.47–11.02	High
PALB2	Breast Cancer Association Consortium, 2021 [118]	26 cases, 8 controls with mutation (<40 years)	5.36	2.26–12.7	High
	Hu et al., 2021 [119]	15 cases, 5 controls with mutation (<45 years)	3.99	2.50–6.67	High
CHD1	Hu et al., 2021 [119]	89 cases, 22 controls with mutation (<45 years)	2.66 *	1.00–8.38	High
TP53	Hu et al., 2021 [119] Breast Cancer Association Consortium, 2021 [118]	NA ^†^	NA ^†^	NA ^†^	High
PTEN	Hu et al., 2021 [119] Breast Cancer Association Consortium, 2021 [118]	NA ^†^	NA ^†^	NA ^†^	High
STK11	Hu et al., 2021 [119] Breast Cancer Association Consortium, 2021 [118]	NA ^†^	NA ^†^	NA ^†^	High
CHEK2	Breast Cancer Association Consortium, 2021 [118]	77 cases, 28 controls with mutation (<40 years)	4.54	2.87–7.17	Moderate
	Hu et al., 2021 [119]	218 cases, 72 controls with mutation (<45 years)	3.06	2.32–4.08	Moderate
ATM	Breast Cancer Association Consortium, 2021 [118]	21 cases, 17 controls with mutation (<40 years)	1.77	0.87–3.59	Moderate
	Hu et al., 2021 [119]	162 cases, 80 controls with mutation (<45 years)	1.89	1.43–2.53	Moderate
BARD1	Breast Cancer Association Consortium, 2021 [118]	6 cases, 3 controls with the mutation (<40 years)	4.30	1.05–17.7	Moderate
	Hu et al., 2021 [119]	33 cases, 22 controls with the mutation (<45 years)	1.37	0.78–2.43	Moderate
RAD51C	Breast Cancer Association Consortium, 2021 [118]	4 cases, 1 control with the mutation (<40 years)	4.83	0.52–45.2	Moderate
	Hu et al., 2021 [119]	26 cases, 20 controls with the mutation (<45 years)	1.26	0.69–2.35	Moderate
RAD51D	Breast Cancer Association Consortium, 2021 [118]	4 cases, 3 controls with the mutation (<40 years)	1.76	0.38–8.17	Moderate
	Hu et al., 2021 [119]	16 cases, 6 controls with the mutation (<45 years)	2.41	0.91–7.60	Moderate
BRIP1	Hu et al., 2021 [119]	41 cases, 35 controls with the mutation (<45 years)	1.22 *	0.75–1.99	Moderate
RAD51B	Hu et al., 2021 [119] Breast Cancer Association Consortium, 2021 [118]	NA ^†^	NA ^†^	NA ^†^	Moderate
XRCC2	Hu et al., 2021 [119]	21 cases, 13 controls with the mutation (<45 years)	1.37 *	0.69–2.83	Moderate
XRCC3	Hu et al., 2021 [119] Breast Cancer Association Consortium, 2021 [118]	NA ^†^	NA ^†^	NA ^†^	Moderate

* Value is not statistically significant (*p* > 0.05). NA ^†^ indicates that age-specific odds ratio was not reported.

**Table 2 cancers-15-01917-t002:** Comparison between lifestyle and reproductive risk factors in pre-menopausal and post-menopausal women.

	Pre-Menopausal Women	Post-Menopausal Women
Lifestyle Risk Factors		
Physical Exercise	Reduces risk of breast cancer	Reduces risk of breast cancer [168]
BMI	Increasing BMI has a modest protective effect against breast cancer	Increasing BMI associated with increased risk of breast cancer
Alcohol Consumption	Increases risk of breast cancer	Increases risk of breast cancer [169]
Smoking	Increases risk of breast cancer, Higher risk in younger age of initiation	Increases risk of breast cancer [105]
Socioeconomic status (SES)	Increasing risk with higher SES	Increasing risk with higher SES
Occupational related long-term night shifts	Increases risk of breast cancer	Increases risk of breast cancer
Reproductive Factors		
Oral contraceptive pills (OCPs)	Increases risk of breast cancer	Increases risk of breast cancer [170]
Levonorgestrel-releasing intrauterine system (LNG-IUS)	Increases risk of breast cancer	Increases risk of breast cancer [171]
Menopausal hormonal therapy (MHT)	NA	Does not increase the risk of breast cancer
Age of pregnancy	Parity before 20 years of age is associated with reduced risk of breast cancer, Parity after 35 years of age is associated with increased risk	Older age of first pregnancy is associated with higher risk of breast cancer [172]
Fertility preservation techniques	Does not increase the risk of breast cancer	Does not increase the risk of breast cancer
Breastfeeding	Reduces risk of breast cancer	Reduces risk of breast cancer

NA indicates “Not Available”.

**Table 3 cancers-15-01917-t003:** Summary of multimodal treatment options for management of breast cancer in young women.

Classification	Treatment	Study, Year	Population	Intervention	Control	Key Findings
Locoregional treatment	Surgery	Vila et al., 2015 [253]	22,598 patients aged ≤40 years	BCS	Mastectomy	No significant difference in risk of death, HR = 0.90 [95%CI: 0.81–1.00]
		Ye et al., 2015 [254]	7665 women aged <40 years	BCS	Mastectomy	No significant differences in 10-year BCSS or 10-year OS 10-year BCSS = 87.7% [95%CI: 86.5–88.9%] for BCS vs. 85.4% [95%CI: 83.8–87.9%] for mastectomy 10-year OS = 85.9% [95% CI: 84.5–87.3%] for BCS vs. 83.5% [95%CI: 81.9–85.1%] for mastectomy
		Li et al., 2022 [255]	1520 patients aged ≤35 years	BCS	Mastectomy	Significantly improved 5-year DFS and 5-year OS for BCS compared to mastectomy 5-year DFS: HR = 0.45 [95% CI: 0.28–0.73], *p* = 0.001 5-year OS: HR = 0.41 [95% CI 0.21–0.80], *p* = 0.009
	Radiation therapy	Bartelink et al., 2015 [256]	449 women aged ≤40 years with stage I and II breast cancer	Whole-breast irradiation (50 Gy in 5 weeks) with 16 Gy boost	Whole-breast irradiation (50 Gy in 5 weeks) alone	Significantly lower 20-year ipsilateral breast tumour recurrence: HR = 0.56 [99% CI 0.34–0.92], *p* = 0.003
		Garg et al., 2007 [257]	107 women aged ≤35 years with stage II or III breast cancer	Mastectomy + postmastectomy radiotherapy (PMRT)	Mastectomy alone	Significantly improved 5-year locoregional control: 88% vs. 63%, *p* = 0.001 Significantly improved 5-year OS: 67% vs. 48%, *p* = 0.03
Systemic treatment	Endocrine therapy	Early Breast Cancer Trialists’ Collaborative Group (EBCTCG), 2005 [258]	2027 women aged <50 years of age with ER+ breast cancer	6 months of anthracycline-based chemotherapy + 5 years adjuvant tamoxifen	6 months of anthracycline-based chemotherapy alone	57% reduction in annual breast cancer mortality rate
		Gray et al., 2013 (aTTOM trial) [259]	6953 women with ER+ or ER-untested BC	10 years of tamoxifen treatment	5 years of tamoxifen treatment	Continued tamoxifen reduced breast cancer recurrence (*p* = 0.003) Recurrence rate ratio (RRR) = 0.99 [95% CI: 0.86–1.15] during years 5–9 RRR = 0.75 [95%CI: 0.66–0.86] at ≥10 years
		Davies et al., 2013 (ATLAS trial) [260]	6846 women with ER+ early BC	10 years of tamoxifen treatment	5 years of tamoxifen treatment	Continued tamoxifen reduced risk of breast cancer recurrence (*p* = 0.002) RRR = 0.90 [95% CI: 0.79–1.02] during years 5–9 RRR = 0.75 [95% CI: 0.62–0.90] ≥10 years Breast cancer mortality RR = 0.97 [95% CI: 0.79–1.18] during years 5–9 Breast cancer mortality RR = 0.71 [95%CI: 0.58–0.88] ≥10 years
		Gnant et al., 2009 [261]	1803 premenopausal women with stage I or II ER/PR+ breast cancer	Anastrozole	Tamoxifen	No significant difference in disease-free survival (DFS): 92.0% vs. 92.8%
		Pagani et al., 2014 (TEXT trial) [262] Francis et al., 2015 (SOFT trial) [263]	4690 premenopausal women with hormone- receptor–positive early breast cancer	5 years of tamoxifen plus ovarian suppression, or exemestane plus ovarian suppression	5 years of tamoxifen	Significantly higher 8-year DFS for exemestane + OFS compared to tamoxifen + OFS group: 86.8% vs. 82.8%, HR (recurrence, a second invasive cancer, or death) = 0.77 [95% CI: 0.67–0.90], *p* < 0.001 Significantly higher 8-year OS for tamoxifen + OS vs. tamoxifen alone: 93.3% vs. 91.5%, *p* = 0.01
		Early Breast Cancer Trialists’ Collaborative Group (EBCTCG), 2022 [264]	7030 premenopausal women with ER+ tumours	Aromatase inhibitors (anastrozole, exemestane, or letrozole) and ovarian suppression	Tamoxifen and ovarian suppression	Lower breast cancer recurrence for women who received AI and ovarian suppression: RR = 0.79 [95% CI 0.69–0.90], *p* = 0.0005 Highest benefit in years 0–4 and no significant difference at >5 years
	Chemotherapy (neoadjuvant)	Huober et al., 2010 [265]	2072 patients with operable or locally advanced breast cancer	6–8 cycles of docetaxel, doxorubicin, and cyclophosphamide (TAC)	2 cycles of TAC followed by 4 cycles of vinorelbine and capecitabine	Highest pathological complete response (pCR) rate in women aged <40 years with TNBC or grade 3 tumours compared to women aged ≥40 years: 57.0% vs. 34.0%, *p* < 0.0001 Women <40 years had double the odds of pCR at surgery compared to those ≥40 years: OR = 2.02 [95% CI: 1.569–2.610] *p* < 0.0001
		Loibl et al., 2015 [266]	1453 breast cancer patients	Neoadjuvant chemotherapy in women aged <40 years	Neoadjuvant chemotherapy in women aged 40–49 and >50	A significantly higher pathological complete remission in the young (< 40 years) group compared with older groups: 20.9 vs. 17.7 vs. 13.7%, *p* < 0.001 Young women with HR+/HER2- and TNBC disease were more likely to achieve pCR after NAC compared to older women: 11 % vs. 5.8 %, *p* < 0.001 in HR+/HER2- and 39.3 % vs. 25.2 %, *p* < 0.001 in TNBC Older women (≥50 years) had significantly better survival outcomes with improved OS (HR = 0.87; 95% CI: 0.74–1.02; *p* = 0.079), LRFS (HR = 0.64; 95% CI: 0.52–0.79; *p* < 0.001), and DFS (HR = 0.81; 95% CI: 0.71–0.92; *p* = 0.001) compared to young women <40 years
		Ahn et al., 2007 [19]	1444 women aged <35 years with breast cancer	NA	NA	Young women were less likely to benefit from adjuvant hormone therapy with higher de novo tamoxifen resistance compared to older women
		Spring et al., 2017 [267]	170 young women aged ≤40 years	Received neoadjuvant chemotherapy and achieved pCR	Received neoadjuvant chemotherapy and did not achieve pCR	Attaining pCR was associated with significantly improved 5-year DFS: 91% vs. 60% for those without pCR, HR = 0.12 [95% CI: 0.04–0.39] *p* < 0.001 Achievement of pCR was associated with significantly improved 5-year OS for patients with pCR compared to those who did not: 95% vs. 75%, HR = 0.19 [95% CI: 0.06–0.62], *p* = 0.006 for all receptor subtypes
		Zujewski et al., 2017 [268]	910 patients	Capecitabine	No additional therapy	Capecitabine had a statistically significant survival advantage compared with no additional therapy
		Mayer et al., 2021 [269]	775 patients with clinical stage II or III TNBC	Platinum agent	Capecitabine	Platinum agents do not improve outcomes in patients with basal subtype TNBC RD post-NAC and are associated with more severe toxicity when compared with capecitabine
		Loibl et al., 2021 [270]	1250 patients with hormone receptor-positive, human epidermal growth factor receptor 2-negative primary breast cancer without a pathological complete response after taxane-containing NACT and at high risk of relapse	Palbociclib	Placebo	Palbociclib for 1 year in addition to ET did not improve iDFS in women with residual invasive disease after NACT
	Chemotherapy (adjuvant)	Sparano et al., 2018 [271]	10,273 women with HR+, HER2-, axillary node-negative breast cancer	Chemoendocrine therapy	Endocrine therapy alone	Adjuvant endocrine therapy and chemoendocrine therapy had similar efficacy Exploratory subgroup analyses of women aged ≤50 years who received chemoendocrine treatment with intermediate RS of 16–25 found a 5-year distant recurrence benefit = 0.8–3.2% and 9-year = 1.6–6.5% No significant difference in overall survival
		Kalinsky et al., 2021 [272]	5015 women with HR+, HER2- breast cancer, one to three positive axillary lymph nodes, and a recurrence score of 25 or lower	Chemoendocrine therapy	Endocrine therapy only	Among premenopausal women, those who received chemoendocrine therapy had longer iDFS and distant relapse-free survival than those who received endocrine-only therapy
		Piccart et al., 2021 [273]	6693 women aged 18–70 years with localized breast cancer (stage T1, T2, or operable T3) with up to three positive lymph nodes	Chemotherapy	No chemotherapy	8-year distant metastasis-free survival rates for those classified as high clinical risk and low genomic risk = 92.0% [95%: CI 89.6–93.8] for the chemotherapy cohort vs. 89.4% [95%CI: 86.8–91.5] for no chemotherapy, HR = 0.66 [95% CI:0. 48–0.92]
		Early Breast Cancer Trialists’ Collaborative Group (EBCTCG) 2022 [264]	7030 premenopausal women with ER+ tumours	Aromatase inhibitors (anastrozole, exemestane, or letrozole) and ovarian suppression	Tamoxifen and ovarian suppression	Anthracycline-based chemotherapy was associated with a larger reduction in the annual breast cancer death rate in younger compared to older women: 38% vs. 20% (independent of hormone-receptor status, tamoxifen use, nodal status, and other tumour features)
		De Laurentiis et al., 2008 [274]	22,903 patients	Taxanes and anthracycline-based chemotherapy	Anthracycline-based chemotherapy	Adding taxanes to anthracycline-based chemotherapy was associated with improved disease-free survival (pooled HR = 0.83; 95% CI: 0.79–0.87; *p* < 0.00001) and overall survival (pooled HR = 0.85; 95% CI: 0.79–0.91; *p* < 0.00001) in both younger and older women with high-risk early stage breast cancer
		Kroman et al., 2000 [15]	10,356 patient, <50 years	NA	NA	No age-specific differences in survival among women who received chemotherapy. However, younger women who did not receive chemotherapy aged <40 years had a higher relative risk of death at 10-ten years of follow-up compared to those aged 45–49 years (RR = 1.40; 95% CI: 1.10–1.78 for ages 35–39 and RR = 2.18; 95% CI: 1.64–2.89 for age <35)
	Targeted therapy	Schmid et al., 2020 [275]	602 patients	Pembrolizumab and chemotherapy	Placebo and chemotherapy	Pembrolizumab (PD-L1 inhibitor) and neoadjuvant chemotherapy was significantly more effective compared to placebo and neoadjuvant chemotherapy in young patients (64.8% vs. 51.2% pCR; *p* < 0.01)
		Loibl et al., 2019 [276]	174 patients	Durvalumab and nab-paclitaxel followed by standard EC	Placebo and nab-paclitaxel followed by standard EC	Durvalumab (PD-L1 inhibitor) and chemotherapy was significantly more effective compared to placebo and chemotherapy
		Bianchini et al., 2020 [277]	280 patients nab-paclitaxel/carbo (CT) or with atezolizumab (CT/A)	Atezolizumab (CT/A)	Nab-paclitaxel/carbo (CT)	No significant difference in the pCR (42.3% for placebo vs. 47.1% for atezolizumab group; *p* = 0.66) in 279 patients with early-stage TNBC randomized to receive neoadjuvant chemotherapy alone versus with atezolizumab for the general population
		Early Breast Cancer Trialists’ Collaborative group (EBCTCG) 2021 [278]	13,864 patients	Trastuzumab and chemotherapy	chemotherapy	The addition of trastuzumab to chemotherapy reduced recurrence rates and breast cancer-related mortality by approximately a third, irrespective of patient age and tumour characteristics
		Earl et al., 2019 [279]	2045 patients	Six-month trastuzumab treatment	Twelve-month trastuzumab treatment	Six-month trastuzumab treatment is not inferior to twelve-month treatment in patients with HER2-positive early breast cancer
		Inno et al., 2019 [280]	11,381 patients	Shorter trastuzumab treatment	Standard trastuzumab treatment	One-year adjuvant trastuzumab is correlated with better DFS and OS compared with shorter durations.
		Piccart et al., 2021 [281]	4805 patients (13.6% of patients aged <40 years in each treatment arm)	1-year pertuzumab added to standard adjuvant chemotherapy and 1-year trastuzumab.	Placebo added to standard adjuvant chemotherapy and 1-year trastuzumab.	Adjuvant pertuzumab in addition to trastuzumab and standard chemotherapy was associated with a significant improvement in the 3-year rate of invasive-disease–free survival compared to trastuzumab and chemotherapy alone (92.0% vs. 90.2%; invasive-disease event HR = 0.77; 95% CI: 0.62–0.96; *p* = 0.02) among patients with HER2+, node-positive breast cancer
		von Minckwitz et al., 2019 [282]	1486 patients (20% of patients aged <40 years in each treatment arm)	Adjuvant Trastuzumab-DM1	Trastuzumab	Significantly improved 3-year invasive disease-free survival by 11.3% in T-DM1 group (invasive disease or death HR of 0.50; 95% CI: 0.39–0.64; *p* < 0.001)
	Metastatic setting	Aebi et al., 2014 [283] Wapnir et al., 2018 [284]	162 patients	Chemotherapy	No chemotherapy	Improved 5-year DFS in women who received chemotherapy for isolated locoregional recurrence (ILRR) after surgical removal with negative margins: 5-year DFS of 69% (95% CI 56–79) with chemotherapy versus 57% (95%CI: 44–67) without chemotherapy (HR = 0.59; 95% CI: 0.35–0.99; *p* = 0.046)
		Michaud et al., 2001 [285]	Meta Analysis of 4 trials involving 464 premenopausal patients	Combined endocrine therapy	Monotherapy	Significant improved median survival (2.9 years with combination vs. 2.5 years with LHRH agonist alone; *p* = 0.02) and median PFS with combined therapy versus monotherapy (8.7 vs. 5.4 months, *p* = 0.0003)
		Carlson et al., 2010 [286]	32 patients	Goserelin plus anastrozole	NA	Goserelin plus anastrozole has substantial antitumour activity in the treatment of premenopausal patients
		Cheung et al., 2005 [287]	36 premenopausal patients with metastatic and locally advanced disease	Goserelin plus anastrozole	NA	The combinations of ovarian function suppression (Goserelin) and Aromatase inhibitors produced sustained clinical benefit and minimal side effects in premenopausal women
		Forward et al., 2004 [288]	16 premenopausal women with metastatic breast cancer or locally advanced primary breast cancer	Goserelin and Anastrozole	NA	The combination of goserelin and anastrozole as second-line endocrine therapy produced a significant clinical response
		Nishimura et al., 2013 [289]	37 premenopausal women with estrogen receptor (ER)-positive and/or progesterone-receptor positive, advanced or recurrent breast cancer refractory to an LH-RH analogue plus tamoxifen	Goserelin and Anastrozole	NA	The combination of goserelin and anastrozole is a safe effective treatment for premenopausal women with hormone receptor-positive, recurrent, or advanced breast cancer
		Torrisi et al., 2007 [290]	32 premenopausal women with T2-T4 N0-N2 breast cancer, whose tumours expressed oestrogen and progesterone receptors.	Letrozole in combination with GnRH analogue	NA	Preoperative letrozole and GnRH analogue are effective in premenopausal women
		Cortes et al., 2011 [291]	762 women with heavily pretreated metastatic breast cancer	Eribulin	Treatment of physician’s choice	Improved overall survival in the eribulin group compared with treatment of physician’s choice (median OS of 13 vs. 11 months; HR = 0.81, 95% CI: 0.66–0.99; *p* = 0.041)
		Schmid et al., 2020 [275]	1174 patients with previously untreated stage II or stage III triple-negative breast cancer	Pembrolizumab, paclitaxel, and carboplatin	Placebo, paclitaxel, and carboplatin	Significantly improved PFS with median PFS of 9.7 months vs. 5.6 months (HR = 0.65; 95% CI: 0.49, 0.86; *p* = 0.0012)
		Robson et al., 2017 [292]	302 patients with a germline BRCA mutation and (HER2)-negative metastatic breast cancer	Olaparib monotherapy	Standard therapy	Superior PFS of olaparib with OS benefit of 7.9 months (22.6 vs. 14.7 months) and a more favourable toxicity profile for patients who did not receive chemotherapy in the metastatic setting
		Litton et al., 2018 [293]	431 patients with advanced breast cancer and a germline BRCA1/2 mutation	Talazoparib	Standard single-agent therapy of the physician’s choice	Superiority of talazoparib with a significantly improved PFS (8.6 months vs. 5.6 months; HR = 0.54; 95% CI: 0.41–0.71; *p* < 0.001) compared to standard single-agent therapy
		Baselga et al., 2012 [294]	808 patients with HER2-positive metastatic breast cancer	Pertuzumab, trastuzumab, and docetaxel	Placebo, and trastuzumab, and docetaxel	Median OS = 56.5 vs. 40.8 months (survival benefit of 15.7 months) in the pertuzumab added group
		Verma et al., 2012 [295]	991 patients with HER2-positive advanced breast cancer, who had previously been treated with trastuzumab and a taxane	Trastuzumab emtansine (T-DM1)	Lapatinib plus capecitabine	OS benefit for T-DM1: 30.9 vs. 25.1 months with less toxicity

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
