# Peer review of "What Is Known about Breast Cancer in Young Women?"

_cancers, 2023, doi:10.3390/cancers15061917_

Round 1

Reviewer 1 Report

     The authors present a very robust review of the management and treatment of breast cancer in young women. It is well written and organized in a clear manner. It is lengthy but is very comprehensive. I have a few minor comments:

·        Please review the paper for consistent position of punctuation and reference number as well as consistent number of spaces in between sentences.

·         How were the papers presented selected?

·         Line 184 – Why do the authors propose are there different findings on the impact of BMI based on race?

·         Line 233 , in stead of .

·         How were the genes selected for representation in Table 1? Based on genes with available studies that have been performed? I recommend including all genes in the table and if no studies have been performed on that particular gene, mention that and include the corresponding penetrance category the gene.

·         Correct line 383 reference punctuation

·         Line 392 change Al to al

·         Line 685 – why do the authors propose that HER2 status did not influence OS where as the MVA discussed in the lines below HER2+ was associated with a poorer OS?

·         Line 786 – Why does the author think that the RCTs of premenopausal women who received chemo or chemoendocrine therapy would have a selection bias of healthier women? Most premenopausal women are healthy enough to receive chemotherapy. Please explain.

·         Line 792 – the word “when” does not make sense in that sentence, recommend removing

·         Section 4.1.1 – did the author find any studies that evaluated lymph node management?

·         Line 994 – Is 402 supposed to be superscript?

·         Line 1012 – Is 286 and 312 supposed to be superscript?

·         Line 1013 – use OFS for consistency with the abbreviation used earlier in the paper

·         Line 1179 – metastatic setting and metastatic breast cancer is redundant, this was also previously abbreviated in the paper, please edit for clarity and consistency

·         Section 5.2 – when discussing surgery, I recommend including that sentinel lymph node evaluation is safe using technetium (not blue dyes)

·         Lines 1272-1277 is rather confusing suggesting that you can delay radiation until after pregnancy but then saying that it isn’t advised due to worse outcomes. Recommend stating that mastectomy is recommended unless the patient is in their third trimester and will deliver prior to causing delays in radiation therapy.

·         Line 1286 – Is this recommendation for duration of endocrine therapy the same for ductal carcinoma in situ and invasive disease?

Author Response

The authors present a very robust review of the management and treatment of breast cancer in young women. It is well written and organized in a clear manner. It is lengthy but is very comprehensive. I have a few minor comments:

Please review the paper for consistent position of punctuation and reference number as well as consistent number of spaces in between sentences.

We have reviewed the paper to ensure consistent formatting including punctuation, reference numbers and spacing between sentences.

How were the papers presented selected?

Our priority was to highlight all available high-quality primary data for young women with breast cancer, both from large primary studies conducted among young women exclusively or data extracted from landmark studies where young women were included as a subgroup analysis. We also tried to include more recent papers wherever possible.

Line 184 – Why do the authors propose are there different findings on the impact of BMI based on race?

The different findings for the impact of BMI based on race may be attributed to several reasons. First, there is a higher rate of surgical menopause in young premenopausal Black women. The higher proportion of premenopausal Black women who undergo hysterectomy with oophorectomy and movement of these women under age 50 subgroup to the postmenopausal subgroup effectively altered the pattern of results in Black women [1]. Secondly, there is a higher prevalence of unfavourable breast cancer tumour characteristics in BIPOC young women, including triple negative breast cancer [2]. Such tumours are less likely to respond to differing estrogen availability associated with obesity and increased BMI. Lastly, there may be genetic differences in the metabolism, estrogen binding and fat distribution among different races.

Line 233 , instead of .

This change has been made.

How were the genes selected for representation in Table 1? Based on genes with available studies that have been performed? I recommend including all genes in the table and if no studies have been performed on that particular gene, mention that and include the corresponding penetrance category the gene.

Changes have been made to table 1 to address this comment. No age specific data was reported for some of the genes which is why they were originally omitted from the table and only mentioned in the body of the manuscript.

Correct line 383 reference punctuation

This change has been made.

Line 392 change Al to al

This change has been made.

Line 685 – why do the authors propose that HER2 status did not influence OS whereas the MVA discussed in the lines below HER2+ was associated with a poorer OS?

We have deleted the “irrespective of HER2 status” to avoid misunderstanding, as this initially was intended to refer to the study comparing survival between ER+ and ER- groups while controlling for other factors including HER2 status. This allows readers to focus on the important point that is subsequently raised, which is that HER+ is associated with significantly poorer OS.

Line 786 – Why does the author think that the RCTs of premenopausal women who received chemo or chemoendocrine therapy would have a selection bias of healthier women? Most premenopausal women are healthy enough to receive chemotherapy. Please explain.

To clarify, we intended to convey that clinical trials would likely underestimate young women with breast cancer who develop metastatic disease given that they may be sicker and hence unable to proceed with the study prior to a formal diagnosis of metastatic disease or lost-to follow-up. We have modified the sentence to reflect this more accurately with “patients with metastatic disease may be sicker and lost to follow-up”

Line 792 – the word “when” does not make sense in that sentence, recommend removing

This change has been made.

Section 4.1.1 – did the author find any studies that evaluated lymph node management?

No, we did not find any high-quality studies examining the management of lymph nodes specifically. 

Line 994 – Is 402 supposed to be superscript?

402 was the previous citation numbering and has now been removed.

Line 1012 – Is 286 and 312 supposed to be superscript?

This change has been made.

Line 1013 – use OFS for consistency with the abbreviation used earlier in the paper

This change has been made.

Line 1179 – metastatic setting and metastatic breast cancer is redundant, this was also previously abbreviated in the paper, please edit for clarity and consistency

This sentence has been edited to avoid redundancy.

Section 5.2 – when discussing surgery, I recommend including that sentinel lymph node evaluation is safe using technetium (not blue dyes)

We have incorporated this point into the manuscript.

Lines 1272-1277 is rather confusing suggesting that you can delay radiation until after pregnancy but then saying that it isn’t advised due to worse outcomes. Recommend stating that mastectomy is recommended unless the patient is in their third trimester and will deliver prior to causing delays in radiation therapy.

Changes have been made to section 5.2.

Line 1286 – Is this recommendation for duration of endocrine therapy the same for ductal carcinoma in situ and invasive disease?

This recommendation is based on expert opinion from the 4th International Consensus Conference for Breast Cancer in Young Women (BCY4) took place in October 2018 by ESO and ESMO. Unfortunately, they did not differentiate between in situ and invasive disease. Sadly, no papers were found that addressed this important distinction with regards to attempting pregnancy after breast cancer. 

the reason we did not include the other genes in the table is that there was no age specific data in the big studies. Or they reported an odds ratio but it was not significant based on P value. I have found smaller studies that report an odds ratio if you want.

Reviewer 2 Report

This manuscript presents an interesting and important review regarding breast cancer in young women. The authors discuss epidemiology including risk factors, prognosis, and treatment. In addition, special considerations including fertility, pregnancy, and bone health are also discussed. The manuscript is informative and interesting. However, there are several points that should be addressed to strengthen the manuscript.   Major comments: 1. In section 2.3.2. Genetic risk factors, and in Table 1, the terms “variant” or “variants” would be better than the terms “mutation” or “mutations”. 2. Please provide information clearly on the differences and similarities between lifestyle and reproductive risk factors for developing breast cancers in young women compared to older women; this could be presented in Tables or Figures. 3. Why is there a higher proportion of HER2-positive breast cancer in young women? Are there any risk factors or specific causes of HER2-positive breast cancer?   Minor comments: 1. Line 16 in the Abstract: “hormone negative tumours” should be “hormone receptor-negative tumours”. 2. Line 53, etc.: “pathogenic mutations” should be “pathogenic variants”. 3. Line 119: “HR+, breast cancer” should be “hormone receptor (HR)-positive breast cancer” or “hormone receptor-positive (HR+) breast cancer” . 4. Line 121: “starting mammography starting” should be “starting mammography screening”. 5. In Figure 1, please provide full terms for “BMI”, “SES”, “OSPs”, “IUDs”, and “SNPs” in the legend. 6. The terms “TNBC” and “triple-negative breast cancer” are both used in the text. Moreover, the terms “endocrine therapy”, “hormonal therapy”, and “ET” are all used. Please use one term consistently and define all abbreviations and acronyms. 7. Lines 1103 and 1110: “HER+” should be “HER2+”.

Author Response

This manuscript presents an interesting and important review regarding breast cancer in young women. The authors discuss epidemiology including risk factors, prognosis, and treatment. In addition, special considerations including fertility, pregnancy, and bone health are also discussed. The manuscript is informative and interesting. However, there are several points that should be addressed to strengthen the manuscript.  

Major comments:

  1. In section 2.3.2. Genetic risk factors, and in Table 1, the terms “variant” or “variants” would be better than the terms “mutation” or “mutations”.

We agree with the proposed change and all “mutation” terms have been replaced with “variant”.

  1. Please provide information clearly on the differences and similarities between lifestyle and reproductive risk factors for developing breast cancers in young women compared to older women; this could be presented in Tables or Figures.

Table 2 was made to address differences and similarities between lifestyle and reproductive risk factors for Pre-menopausal Women and post-menopausal women.

  1. Why is there a higher proportion of HER2-positive breast cancer in young women? Are there any risk factors or specific causes of HER2-positive breast cancer?  

There are no robust studies to explain the higher proportion of HER2+ breast cancer in young women currently. Existing literature suggests that biologic differences including different breast stroma, somatic mutations including TP53 and PIK3CA point mutations and changes that occur during or shortly after pregnancy may influence tumour subtype in young women. [4-6] A previous study with 156 women found that a self-reported history of benign breast disease and the occurrence of HER2 positive breast cancer (OR = 2.1; 95% CI, 1.1-4.1) and no association with family history of breast cancer, parity, late age at first birth, breastfeeding or oral contraceptive use. [7]

Minor comments:

  1. Line 16 in the Abstract: “hormone negative tumours” should be “hormone receptor-negative tumours”.

This change has been made.

  1. Line 53, etc.: “pathogenic mutations” should be “pathogenic variants”.

This change has been made.

  1. Line 119: “HR+, breast cancer” should be “hormone receptor (HR)-positive breast cancer” or “hormone receptor-positive (HR+) breast cancer”

This change has been made.

  1. Line 121: “starting mammography starting” should be “starting mammography screening”.
  2. In Figure 1, please provide full terms for “BMI”, “SES”, “OSPs”, “IUDs”, and “SNPs” in the legend.

We have revised Figure 1 to minimize abbreviations and have included the full terms for all remaining abbreviations in the legend.

  1. The terms “TNBC” and “triple-negative breast cancer” are both used in the text. Moreover, the terms “endocrine therapy”, “hormonal therapy”, and “ET” are all used. Please use one term consistently and define all abbreviations and acronyms.

We have edited the terms to ensure consistent terminology is used throughout the manuscript.

  1. Lines 1103 and 1110: “HER+” should be “HER2+”.

These changes have been made.

References

[1] Hall IJ, Newman B, Millikan RC, Moorman PG. Body size and breast cancer risk in black women and white women: the Carolina Breast Cancer Study. Am J Epidemiol. 2000;151(8):754-764. doi:10.1093/oxfordjournals.aje.a010275.

[2] Siddharth S, Sharma D. Racial Disparity and Triple-Negative Breast Cancer in African-American Women: A Multifaceted Affair between Obesity, Biology, and Socioeconomic Determinants. Cancers (Basel). 2018;10(12):514. Published 2018 Dec 14. doi:10.3390/cancers10120514.

[3] Purrington KS, Gorski D, Simon MS, et al. Racial differences in estrogen receptor staining levels and implications for treatment and survival among estrogen receptor positive, HER2-negative invasive breast cancers. Breast Cancer Res Treat. 2020;181(1):145-154. doi:10.1007/s10549-020-05607-4.

[4] Stephens P.J., Tarpey P.S., Davies H., et al. The landscape of cancer genes and mutational processes in breast cancer. Nature. May 16 2012;486(7403):400–404. doi: 10.1038/nature11017.

[5] Shah S.P., Roth A., Goya R., et al. The clonal and mutational evolution spectrum of primary triple-negative breast cancers. Nature. Apr 4 2012;486(7403):395–399. doi: 10.1038/nature10933.

[6] Kim HJ, Kim S, Freedman RA, Partridge AH. The impact of young age at diagnosis (age <40 years) on prognosis varies by breast cancer subtype: A U.S. SEER database analysis. Breast. 2022;61:77-83. doi:10.1016/j.breast.2021.12.006.

[7] Swede H, Moysich KB, Freudenheim JL, et al. Breast cancer risk factors and HER2 over-expression in tumors. Cancer Detect Prev. 2001;25(6):511-519.